# Preoperative admission is non-essential in most patients receiving elective laparoscopic cholecystectomy: A cohort study

Suppadech Tunruttanakul[1]*, Ratchanee Tunruttanakul[2], Kamoltip Prasopsuk[3], Kwanhathai Sakulsansern[1], Kyrhatii Trikhirhisthit[4]

1 Department of Surgery, Sawanpracharak Hospital, Nakhon Sawan, Thailand, 2 Nong pling, Muang, Nakhon Sawan, Thailand, 3 Department of Anesthesiology, Regional Health Promotion Center 3, Nakhon Sawan, Thailand, 4 Department of Radiology, Division of Radiation Oncology, Sawanpracharak Hospital, Nakhon Sawan, Thailand

* suppadech_t@spr.go.th

**Data Availability Statement:** All relevant data are within the paper and its Supporting Information files.

## Abstract

We evaluated conventional overnight-stay laparoscopic cholecystectomy, focusing on the preoperative admission day, to assess the feasibility of implementing daycare laparoscopic cholecystectomy, which is currently underutilized in developing and some Asian countries. We retrospectively reviewed elective laparoscopic cholecystectomy data from March 2020 to February 2022 at a 700-bed tertiary hospital in Thailand. Variables included age, sex, body mass index, comorbidities, American Society of Anesthesiologists status, presence of preoperative anesthesiology visit, laparoscopic cholecystectomy indications, additional intraoperative cholangiography, and surgery cancellations. The primary focus was on preoperative treatment and monitoring needs; secondary outcomes included morbidity, mortality within 30 days, and prolonged hospital stay (>48 hours). Statistical analysis was conducted using the Fisher exact test, $t$-test, and logistic regression. The study included 405 patients. Of these, 65 (16.1%) received preoperative treatment, with 21 unnecessary (over) treatments and six under-treatments. Based on the results, approximately 12.1% (n = 49) of patients may have theoretically required preoperative admission and treatment. Multivariable analysis showed that the increasing of comorbidities was significantly associated with preoperative management (odds ratio [95% Confidence interval]: 7.0 [2.1, 23.1], 23.9 [6.6, 86.6], 105.5 [17.5, 636.6]) for one, two, and three comorbidities, respectively), but factors such as age, obesity, and American Society of Anesthesiologists status were not. The cohort had 4.2% morbidity (2.2% medical complications), with no mortality. Surgery cancellations occurred in 0.5%. In conclusion, on the basis of our data, a small proportion (12.1%) of patients undergoing elective laparoscopic cholecystectomy may require preoperative admissions to receive the necessary treatment, and most (87.9%) preoperative admissions may not provide treatment benefit. The traditional admission approach was safe but required re-evaluation for optimal resource management.

**Funding:** This study received financial support from Sawanpracharak Hospital Medical Education center (https://mec.spr.go.th). The funders had no role in study design, data collection and analysis, decision to publish, or preparation of the manuscript.

**Competing interests:** All authors declare that there is no conflict of interest.

# Introduction

Laparoscopic cholecystectomy (LC) is a frequently performed procedure [1] used to treat various complications related to gallstones [2]. With advancements in perioperative care, many less-invasive surgeries, including LC, can be performed without the need for a conventional overnight hospital stay [3]. In Western countries, practices have shifted toward ambulatory or daycare LC, where patients are able to return home on the same day as their surgery [4,5]. However, daycare surgery has yet to gain widespread acceptance in Asia, particularly in developing countries [6–8].

The Thai Ministry of Public Health has launched an initiative to promote daycare surgery, but it previously only covered procedures requiring local anesthesia or sedation. In 2021, LC was added to the list of eligible procedures. Given the limited experience with daycare surgery in Thailand, we reviewed our conventional LC with overnight hospital stay. Since the traditional overnight-stay approach involves a significant proportion of time spent on preoperative admission, we aimed to determine whether first-day admission was necessary and which patients would benefit from this approach by focusing specifically on the preoperative admission day.

# Material and methods

The study design followed a retrospective observational cohort approach and involved a review of data from all patients who underwent elective LC over 2 years, from March 2020 to February 2022. The study data collection and access for research purposes began in mid-December 2022 and ended in February 2023. Individuals under 18 years old, emergency LC cases, and incomplete records were excluded from the analyzed data. Because our focus was on patients undergoing elective LC, we excluded cases involving more complex adjunctive procedures such as laparoscopic bile duct exploration and same-admission preoperative endoscopic retrograde cholangiography. However, we included patients receiving elective LC who underwent intraoperative cholangiography (IOC) because it is a relatively simple procedure with minimal additional time requirements [9]. Furthermore, many countries have adopted a routine approach to IOC for all patients undergoing LC [10].

The setting was Sawanpracharak Hospital, a 700-bed tertiary hospital located in Nakhon Sawan, Thailand. Most of the patient data analyzed was limited to this region because LC are typically available in local provincial hospitals, and referral to other facilities is unnecessary. All patient data were obtained from the hospital's electronic medical records system. We followed Helsinki Declaration's confidentiality principles and kept all participants anonymous. No individual data was used in our analysis. The study protocol was approved by the Sawanpracharak Hospital Ethical Committee for Research in Human Subjects (COA.39/2022). The requirement for patient consent was waived because of the retrospective study design and use of deidentified data.

In this study, we reviewed the following parameters that could potentially impact the study objective: patient's age, sex, body mass index (BMI), American Society of Anesthesiologists (ASA) status, comorbidities, presence of preoperative anesthesiology clinic (PAC) visits, LC indications, presence of additional IOC, and surgery cancellations. The patient's age was recorded in years and grouped into two categories: age$\leq$ 65 years old or age$>$ 65 years old (according to the cutoff in the review by Orimo et al. [11]). BMI was recorded in kg/m$^2$ and divided into three categories based on the Asian obesity cutoff value for public health action: $<27.5$ kg/m$^2$, 27.5 to $<32.5$, and $\geq32.5$ kg/m$^2$ [12]. ASA status was determined by anesthesiologists. Comorbidities included anemic diseases, diabetes mellitus, hypertension, asthma or chronic obstructive pulmonary disease (COPD), and heart disease (confirmed by

cardiologists). Heart diseases that can affect surgery include coronary artery diseases, arrhythmias, ventricular dysfunction, or congestive heart failure [13].

We included various gallstone-related indications for elective LC, encompassing symptomatic gallstones, prior acute cholecystitis, prior gallstone pancreatitis, and prior choledocholithiasis. For cases of acute cholecystitis or gallstone pancreatitis, we only included patients who showed symptom improvement or resolution after initial conservative treatment and who subsequently underwent elective LC. Patients who underwent early LC during the first episode of acute cholecystitis or gallstone pancreatitis, which were considered emergency cases, were excluded from the study. Additionally, patients with recurrent biliary admissions for either condition that necessitated urgent LC before the scheduled procedure were also excluded. Choledocholithiasis included only patients who had previously undergone endoscopic retrograde cholangiography to remove their stones and were scheduled for elective LC. We maintained data heterogeneity to address the diverse indications commonly seen in elective LC, ensuring the relevance and applicability of our study.

Regarding the presence of IOC parameters, in our department, selective IOC was indicated for patients with symptomatic gallstones who had a relatively low but non-negligible risk of choledocholithiasis. This included patients with a clinical history of jaundice, pancreatitis, abnormal liver function test results, bile duct dilatation detected in imaging studies, or the patients at intermediate risk for choledocholithiasis according to guidelines [14]. In cases with a greater suspicion of choledocholithiasis (high-risk for choledocholithiasis, which includes clinical cholangitis or choledocholithiasis detected in imaging studies), alternative modalities such as endoscopic retrograde cholangiography were used [14]. Choledocholithiasis detected by IOC was managed based on the decision of the attending surgeon. Treatment options included transcystic bile duct stent placement [15] or performing endoscopic retrograde cholangiography during the same procedure [16].

PAC visits were conducted for all patients undergoing elective surgeries at our hospital as outpatient consultations and took place on the same day if surgeons and patients agreed to proceed with planned surgeries. Anesthesiologists, along with nurse anesthetists, conducted all pre-anesthetic evaluations. Additional consultations with subspecialists could be arranged if the anesthesiologist required more specific evaluations. Occasionally, the absence of PAC visits in our hospital could occur owing to human error, either on the part of patients or the administrative staff. To ensure comprehensive assessments, in-patient preoperative anesthesiology visits and checklists were routinely completed on the day of preoperative admission. This allowed for further patient evaluation and the execution of necessary treatments. In cases where patients missed PAC visits, they would still undergo in-patient preoperative anesthesiology visits to prevent cancellations. However, surgical cancellations could still occur if patients were deemed unfit for the planned operations. All surgeries had to be approved by the anesthesiologists before the operation could be performed. Preoperative screening tests were conducted during both the PAC visits and in-patient preoperative anesthesiology visits. Chest radiography and a complete blood count (blood test that evaluates the total numbers and characteristics of cell components in a patient's blood, including red and white blood cells and platelets) were performed for all patients. Patients over 45 years old underwent electrocardiogram testing, and those over 60 years old had blood urea nitrogen, creatinine, and electrolyte levels further assessed. Our hospital's preoperative screening test protocol with respect to patients' age (i.e., blood tests, chest radiography, and electrocardiogram testing) has been adopted from the practice advisory for pre-anesthesia evaluation by the Royal College of Anesthesiologists of Thailand [17]. Additional laboratory tests were organized based on specific comorbidities, such as blood sugar testing for diabetes mellitus. Liver function tests were conducted for all patients undergoing LC.

The primary objective of this study was to evaluate the treatment and monitoring received by patients during their preoperative admission, excluding care provided on the day of surgery. Two participating specialists, a surgeon and an anesthesiologist, separately reviewed the patient data and later held an in-person meeting to review their findings. In cases of inconclusive results, a subspecialist in internal medicine was consulted. The criteria for necessary preoperative treatment are outlined in S1 Table. Based on the described process, preoperative treatment was classified into three categories: "treatment received", over-treatment, and under-treatment. Subsequently, the category of "treatment necessary", representing necessary treatment for patients, was determined based on the aforementioned categories. The details of each category are as follows:

- "Treatment received" referred to treatments that were provided and documented in the patient's records.

- Over-treatment denoted treatment that should not have been given.

- Under-treatment referred to treatment that was necessary but not given.

- "Treatment necessary" referred to "treatment received", excluding cases of over-treatment and including cases of under-treatment after exclusion.

In our analysis, each patient was treated as an individual unit for evaluation. Including under-treatment after excluding over-treatment was intended to prevent the exclusion of patients who experienced both under-treatment and over-treatment. "Treatment necessary" comprised all patients who received the preoperative treatment (i.e., the "treatment received" group), as well as those who required treatment but did not receive it or received insufficient treatment (i.e., the under-treatment group). However, "treatment necessary" excluded patients in the over-treatment group. Only surgery cancellations owing to medical reasons [18] were considered "treatment necessary". The same criteria used to evaluate preoperative treatment were applied to patients with other reasons for cancellation. All necessary preoperative cardiovascular instability monitoring was also considered preoperative treatment (S1 Table). Monitoring preoperative cardiovascular instability is a standard protocol at our hospital for high-risk non-cardiovascular surgical patients. On the preoperative day, anesthesiologists require an extended monitoring period, which includes non-invasive blood pressure and vital sign measurement, continuous electrocardiographic monitoring, and peripheral pulse oximetry measurements [19]. Additionally, in-patient preoperative anesthesiology visits, checklists, and a review of PAC assessments are conducted. Additional monitoring during the operative and postoperative periods is also implemented but beyond the scope of this study. This process is designed to enhance the safety of surgery for patients deemed at high risk for perioperative complications (S1 Table).

Secondary outcomes of the study were postoperative morbidity and mortality that occurred within 30 days after surgery and the length of hospital stay (LOHS). Complications were classified according to Clavien–Dindo classification [20].

Patient discharge were determined by the attending surgeon. In general, the criteria for discharge from our department included stable vital signs, normal breathing with oxygen saturation levels above 90% on room air, reasonable mobility, and a manageable level of pain, nausea, and vomiting. Regarding LOHS, considering the expected positive skewness of LOHS and the focus of our study on evaluating the feasibility of daycare surgery, we categorized LOHS into two groups: ≤48 hours and >48 hours. LOHS exceeding 48 hours was considered a prolonged stay, equivalent to a postoperative stay of more than 24 hours (excluding the 24-hour preoperative admission period) [21].

Our surgical department routinely conducts a 1-month follow-up. Assessment of clinical symptoms, physical examinations, and reviews of histologic reports are performed at outpatient visits during follow-up for patients undergoing LC. Additionally, liver function tests are used to monitor patients undergoing LC plus IOC. In the event that patients were lost to follow-up, they were contacted by phone to inquire about postoperative complications or were admitted to other hospitals during the 30-day postoperative period. We planned to conduct a complete case analysis to address any missing data.

Our study size was determined using the number of events per variable method [22], which requires 10 events per variable in the main outcome analysis using logistic regression. Based on the historical records of 50 patients, we found that approximately 18% of elective LC cases may require preoperative treatment. To achieve 70 events with seven variables included (age, sex, BMI, ASA status, comorbidities, LC indications, and additional procedures [IOC]), the study size was calculated to be 389 patients, considering an expected preoperative treatment rate of 0.18.

Regarding data analysis, categorical parameters were compared using the Fisher's exact test, and continuous data were compared using a *t*-test. Multivariable analysis was performed using logistic regression to examine the associations between multiple potential predictors and preoperative treatment, complication outcomes, and prolonged LOHS (>48 hours). Predictors included in the multivariable analysis are outlined in Table 3. The goodness of fit of the logistic regression models was tested using the Hosmer–Lemeshow test. This assessment involves grouping the observations into deciles (groups of 10) based on their predicted probabilities [23]. Finally, sensitivity analysis was conducted to investigate outcome variability according to alteration of LC indications and the presence of additional IOC. A p-value of less than 0.050 was considered statistically significant. Statistical analysis was performed using STATA statistical software, version 17 (StataCorp, College Station, TX, USA). This work has been reported in line with the STROBE criteria [24].

## Results

The study participant flow is illustrated in Fig 1. We collected records of 467 patients who underwent LC during the study period. The records of 62 patients were excluded owing to being under the age of 18 years, having undergone emergency surgery, and having laparoscopic bile duct exploration or same-admission preoperative endoscopic retrograde cholangiography. There was no data loss. Overall, a total of 405 patients were included in the analysis. Among these, twenty-one (5.2%) patients did not complete a 1-month follow-up. Of these, 16 patients were reached by phone, and five (1.2%) were lost to follow-up and could not be contacted.

The patient characteristics are listed in Table 1. The average age was 54.1 (±15.7) years, with a range from 18 to 94 years old. 25.7% were obese with BMI 27.5 to <32.5 kg/m2, and 9.4% were BMI ≥32.5. Most patients had ASA status II (50.4%), with 33.3% having ASA status III. The most common indication for LC was symptomatic gallstones (72.8%). IOC was performed in 6.9% of LCs. Surgery cancellations related to the study protocol were observed in two cases (0.5%). One cancellation was for a medical reason (recently treated hyperthyroidism), and the other was attributed to the unavailability of theater time. The patient who canceled surgery for medical reasons later refused to undergo the procedure. A total of 110 patients (27.2%) experienced a prolonged LOHS (>48 hours). No missing data was identified.

Table 2 presents preoperative treatment information. Sixty-five patients (16.1%) received preoperative treatment ("treatment received"), with 21 (5.2%) being over-treatments and six (1.5%) being under-treatments. One patient received treatment, but the treatment they

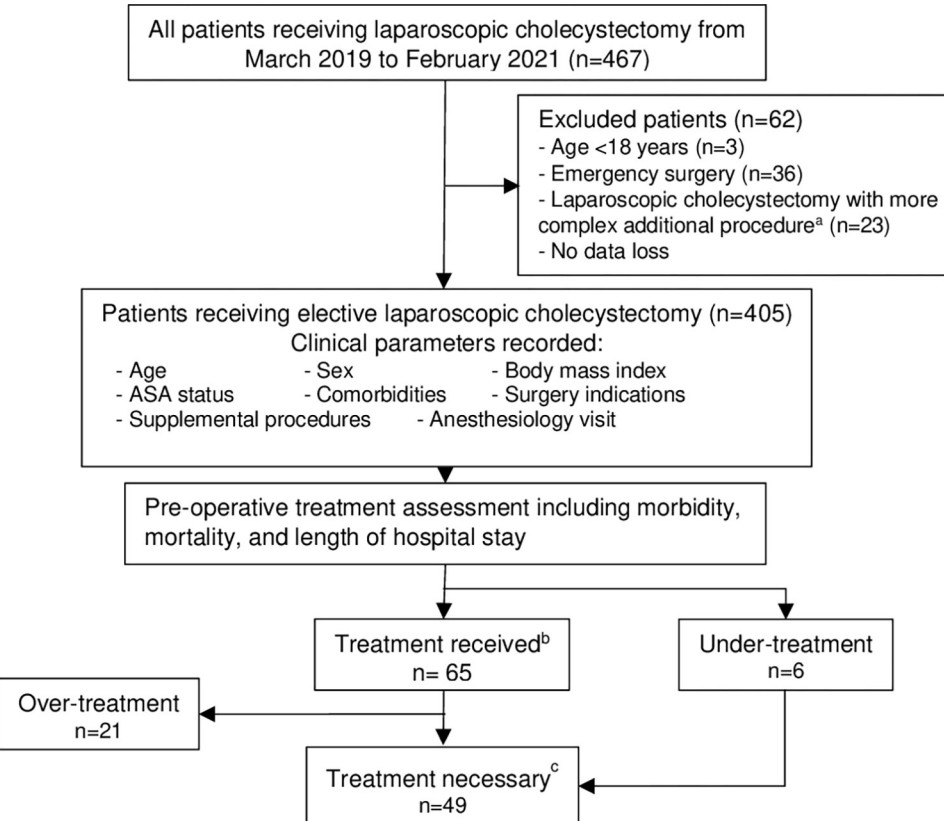

**Fig 1. Study participant flow diagram.** [a]More complex additional procedure: Laparoscopic bile duct exploration and same-admission preoperative endoscopic retrograde cholangiography. [b]Treatment received: Preoperative treatment that was documented in the records. [c]Treatment necessary: Received, excluding over-treatment and including under-treatment cases after exclusion. ASA, American Society of Anesthesiologists.

received was inadequate (under treatment). Based on these findings, 49 (12.1%) patients theoretically required preoperative treatment or monitoring ("treatment necessary"). The most common "treatment received" were potassium corrections (35 patients, 53.8%) followed by monitoring of cardiovascular instability (13 patients, 20%). In the "treatment necessary" group (which includes under-treatments and excludes over-treatments from the "treatment received"), the treatment proportions were relatively similar to the "treatment received". However, potassium correction was lower at 40.8% (20 patients), and monitoring of cardiovascular instability was higher at 26.5% (13 patients). Treatments were found to be associated with most comorbidities, including ASA status, except for asthma or COPD. Notably, in our dataset, patients with ASA status I did not receive preoperative treatment ('treatment received') or need necessary treatment ("treatment necessary") on the preoperative admission date.

Table 3 summarizes the results of multivariable analysis, taking multiple predictors into account. The patient whose surgery was canceled refused to undergo the procedure, resulting in a decrease of one in the number of patients analyzed for medical complications and prolonged LOHS (n = 404 instead of the total n = 405). The analysis showed that the presence of comorbidities was the only significant factor affecting the need for preoperative treatment. The odds of preoperative treatment increased with the number of comorbidities: odds ratio (95% Confidence interval) 7.0 (2.1, 23.1), 23.9 (6.6, 86.6), 105.5 (17.5, 636.6) for one, two, and three comorbidities, respectively. Patients' age, BMI greater than 32.5 kg/m$^2$, presence of IOC,

**Table 1. Participant characteristics.**

| Patient characteristic, n = 405 | n (%) |
|---|---|
| Age (years), mean [±SD] | 54.1 [15.7] |
| ≥65 years | 104 (25.7) |
| Female sex | 312 (77.0) |
| Body mass index (kg/m$^2$) (mean [±SD]) | 25.9 [5.2] |
| BMI 27.5 to <32.5 | 104 (25.7) |
| BMI ≥32.5 | 38 (9.4) |
| ASA status | |
| I | 66 (16.3) |
| II | 204 (50.4) |
| III | 135 (33.3) |
| Comorbidity | 205 (50.6) |
| Two or more comorbid diseases | 63 (15.6) |
| History of anemia | 15 (3.7) |
| Diabetes mellitus | 62 (15.3) |
| Hypertension | 168 (41.5) |
| Asthma or COPD | 13 (3.2) |
| Heart disease | 14 (3.5) |
| Cholecystectomy indications | |
| Symptomatic gallstones | 295 (72.8) |
| Gallstone pancreatitis | 20 (4.9) |
| Common bile duct stones | 45 (11.1) |
| Subsided acute cholecystitis | 45 (11.1) |
| Intraoperative cholangiography | 28 (6.9) |
| Trans-cystic duct biliary stent[a] | 2 (0.5) |
| Same-setting ERC[a] | 2 (0.5) |
| Surgery cancellations<br>Medical reasons | 2 (0.5)<br>1 (0.3) |
| Out-patient pre-anesthesia clinic visit | 340 (84.0) |
| Prolonged hospital stays (> 48 hours) | 110 (27.2) |

[a]Two options were available at the study hospital for managing choledocholithiasis detected during intraoperative cholangiography: Trans-cystic duct biliary stent or same-setting ERC.

ASA: American Society of Anesthesiologists, BMI: Body mass index, COPD: Chronic obstructive pulmonary disease, ERC: Endoscopic retrograde cholangiography, SD: Standard deviation.

surgical indications, and ASA status were not associated with preoperative treatments. Complications occurred in 17 (4.2%) patients, with no mortality; 58.8% of complications were surgery-related (10 patients, 2.8%), including two cases of major morbidity (grade IIIb, bile duct injury requiring surgical reconstruction). Medical complications were experienced by nine patients (2.2%), with two severe cases (grade IV, 0.5%) including one cerebrovascular infarction and one readmission for lobar pneumonia with respiratory failure. Five patients had grade II complications. Three patients (0.7%) experienced retained common bile duct stones. With our sample size, no variable was found to significantly predict medical complications.

The median hospital stay was 3 (interquartile range: 2–4) days. Regarding Table 3, factors that predicted prolonged LOHS (>48 hours) included the presence of IOC, surgical indications other than symptomatic gallstones, patients requiring preoperative treatment, and the occurrence of complications.

**Table 2. Study details regarding preoperative treatment.**

| | Preoperative treatment, n (%)[a] | | | |
|---|---|---|---|---|
| | Treatment received[c], 65 (16.1) | | Treatment necessary[d], 49 (12.1) | |
| Over-treatment | 21 (5.2) | | | |
| Under-treatment | 6 (1.5) | | | |
| Treatment details | | | | |
| Blood component matching or transfusion | 10 (2.5) | | 7 (1.7) | |
| Blood sugar control | 10 (2.5) | | 8 (2.0) | |
| Hypertension control | 6 (1.5) | | 6 (1.5) | |
| Electrolyte correction | | | | |
| Potassium | 35 (8.6) | | 20 (5.0) | |
| Sodium | 2 (0.5) | | 1 (0.3) | |
| Anticoagulant bridging | 3 (0.7) | | 3 (0.7) | |
| Cardiovascular instability monitoring | 13 (3.2) | | 13 (3.2) | |
| Medical reasons for surgery cancellation | 1 (0.3) | | 1 (0.3) | |
| **Treatments regarding conditions** | n (%)[b] | *p*-value | n (%)[b] | *p*-value |
| Treatments regarding ASA status | | <0.001 | | <0.001 |
| I (n = 66) | 0 (0) | | 0 (0) | |
| II (n = 204) | 29 (14.2) | | 18 (8.8) | |
| III (n = 135) | 35 (26.7) | | 31 (23.0) | |
| Treatments regarding comorbidities | | | | |
| BMI ≥32.5 (n = 38) | 9 (23.7) | 0.171 | 9 (23.7) | 0.033 |
| Known history of anemia (n = 15) | 6 (40.0) | 0.021 | 5 (33.3) | 0.025 |
| Diabetes mellitus (n = 62) | 25 (40.3) | <0.001 | 22 (35.5) | <0.001 |
| Hypertension (n = 168) | 47 (28.0) | <0.001 | 43 (25.6) | <0.001 |
| Asthma or COPD (n = 13) | 3 (23.1) | 0.447 | 2 (15.4) | 0.663 |
| Heart disease (n = 14) | 7 (50.0) | 0.003 | 7 (50.0) | <0.001 |

[a]Percentage according to total study population (405 patients).

[b]Percentage of patients in a specific group or with a particular condition.

[c]Treatment received: Preoperative treatment that was documented in the records.

[d]Treatment necessary: Treatment received, excluding cases of over-treatment and including cases of under-treatment after exclusion.

ASA: American Society of Anesthesiologists, BMI: Body mass index, COPD: Chronic obstructive pulmonary disease.

A goodness of fit test (model calibration) was conducted for logistic regression models, assessing the agreement between predicted values from the model and observed values. A non-significant result was interpreted as indicative of good calibration [23]. Regarding models in Table 3, the treatment necessary model had a *p*-value of 0.221, the medical complication model had a *p*-value of 0.138, and the prolonged LOHS model had a *p*-value of 0.126. These results suggest that all models had a proper ability to describe the response variable.

Sensitivity analyses were conducted to explore the impact of more homogeneous data on the main outcome ("treatment necessary"); the results are summarized in S2 Table. When only patients with symptomatic gallstones were included (excluding other indications for LC, n = 295), and when only elective LC without additional procedures was examined (excluding patients who underwent IOC, n = 377), the findings were consistent with the full model. In both sensitivity analyses, the presence of comorbidities was the only significant factor associated with the need for preoperative treatment. The odds of requiring preoperative treatment increased with the number of comorbidities: odds ratios (95% confidence intervals) were 10.8 (2.2, 53.2), 30.9 (5.5, 174.7), and 105.1 (12.8, 863.5) for one, two, and three comorbidities,

**Table 3. Multivariable analysis of predictors for preoperative treatment requirement, medical complications, and length of hospital stay.**

| | Treatment necessary[a] | | | Medical complication | | | Prolonged hospital stays (> 48 hours) | | |
|---|---|---|---|---|---|---|---|---|---|
| | OR (95% CI) | SE | *p*-value | OR (95% CI) | SE | *p*-value | OR (95% CI) | SE | *p*-value |
| Age | 1.0 (1.0, 1.0) | 0.02 | 0.952 | 1.0 (1.0, 1.1) | 0.03 | 0.472 | 1.0 (1.0, 1.0) | 0.01 | 0.946 |
| ≥65 years | 1.1 (0.5, 2.4) | 0.43 | 0.885 | 0.6 (0.1, 3.2) | 0.55 | 0.548 | 1.6 (0.9, 3.0) | 0.50 | 0.118 |
| Female sex | 0.8 (0.4, 1.8) | 0.32 | 0.635 | 0.7 (0.2, 3.2) | 0.54 | 0.669 | 0.6 (0.3, 1.0) | 0.16 | 0.053 |
| BMI ≥32.5 kg/m$^2$ | 1.9 (0.7, 5.5) | 1.03 | 0.213 | 1.6 (0.2, 16.6) | 1.92 | 0.686 | 1.2 (0.5, 2.8) | 0.52 | 0.660 |
| Intraoperative cholangiography | 1.1 (0.3, 4.4) | 0.78 | 0.916 | NA | | | 5.8 (2.4, 14.1) | 2.62 | <0.001 |
| Other surgical indications[b] | 1.1 (0.8, 1.5) | 0.19 | 0.670 | 1.2 (0.7, 2.2) | 0.37 | 0.503 | 1.5 (1.2, 1.9) | 0.16 | <0.001 |
| ASA status III | 1.2 (0.6, 2.7) | 0.50 | 0.602 | 2.4 (0.4, 13.8) | 2.15 | 0.319 | 1.3 (0.7, 2.3) | 0.40 | 0.473 |
| Comorbidity | | | | 1.9 (0.3, 11.7) | 1.75 | 0.506 | 1.0 (0.6, 1.8) | 0.31 | 0.941 |
| One comorbidity | 7.0 (2.1, 23.1) | 4.30 | 0.001 | | | | | | |
| Two comorbidities | 23.9 (6.6, 86.6) | 15.69 | <0.001 | | | | | | |
| Three comorbidities | 105.5 (17.5, 636.6) | 96.73 | <0.001 | | | | | | |
| Treatment necessary[a] | | | | 0.5 (0.1, 4.0) | 0.51 | 0.485 | 2.7 (1.3, 5.4) | 0.97 | 0.007 |
| All complications | | | | | | | 3.0 (1.1, 8.5) | 1.60 | 0.040 |

n = 405 for treatment necessary outcome, n = 404 for medical complication outcome, and n = 404 for prolonged hospital stay outcome. One patient who had a surgery cancellation refused to undergo surgery and was excluded from the analysis.

NA: Dataset does not have enough information to reasonably estimate the effect secondary to low incidence of complications (n [%], 9 [2.1%]).

[a]Treatment necessary: Preoperative treatment that was documented in the records, excluding cases of over-treatment and including cases of under-treatment after exclusion.

[b]Other surgical indications: Indications for laparoscopic cholecystectomy other than symptomatic gallstones.

ASA: American Society of Anesthesiologists, BMI: Body mass index, CI: Confidence Interval, OR: Odds ratio, SE: Standard error.

respectively, in the model with symptomatic gallstones as the sole indication. Similarly, in the model without additional IOC, the odds ratios (95% confidence intervals) were 8.5 (2.3, 32.4), 25.1 (6.0, 105.7), and 118.5 (17.9, 786.2) for one, two, and three comorbidities, respectively. Other factors were not found to be associated with preoperative treatment in either of the sensitivity analytic models.

## Discussion

LOHS is an essential indicator of hospital management. A shorter LOHS has several benefits, such as a reduced risk of infection, decreased medication side effects, improved treatment quality, and increased hospital profits [25]. Because Thailand has national universal health care coverage [26], the financial advantage of reducing LOHS may be reduced. However, disregarding LOHS has resulted in an overburdening of case numbers in many hospitals, particularly tertiary hospitals. Our tertiary hospital also faces a problem of overcrowding in its inpatient department. In general, hospital beds cannot be limited according to policies concerning government hospitals. To address this issue, reducing overcrowding is a priority for the Ministry of Public Health. Our findings indicated that a significant proportion (87.9% [regarding the "treatment necessary" group]) of patients with elective preoperative LC admissions did not receive beneficial treatment. Patients with lower ASA status demonstrated a lower rate of preoperative treatment, with no patients who had ASA status I requiring treatment and only 8.9% of those with ASA status II having the necessary treatment. Moreover, some patients were at risk of receiving over-treatment (21 [32.3%] out of the 65 in the "treatment received"). These results suggest that our limited resources may have been misused in caring for patients unnecessarily.

Our data implied that in terms of patient factors, age, obesity, and ASA III were not associated with the need for preoperative treatment, after accounting for multiple predictors in the multivariable analysis (Table 3). However, the number of comorbidities was found to be significantly related to treatment requirements. Our results are in line with those of previous studies suggesting that hospital admission may not always be necessary for elderly and obesity patients: however, these factors are often associated with comorbidities that can impact surgical considerations [27–29]. Whereas ASA III status refers to a patient with a non-life-threatening severe systemic disease and is always associated with comorbidities, this may explain why it was significantly related to treatment requirements in univariable analysis but not in the multivariable analysis. Some studies also support performing daycare surgery in selected ASA III patients with stable systemic diseases [30,31].

The preoperative treatment in this study (Table 2) showed that potassium correction was the most frequently administered treatment (53.8%, 35 out of 65 patients). Unfortunately, this was also the primary cause of over-treatment, with 15 out of 21 over-treatments (71.4%) being attributed to potassium correction. Our data indicated that most cases of hypokalemia were owing to factors such as diuretic drugs and poor oral intake. However, when hypokalemia was detected incidentally in routine preoperative tests, it was often negligible (>3.0 mEq) and resulted in unnecessary treatment. Recent guidelines from the United Kingdom (National Institute for Health and Care Excellence) have shifted away from recommending routine preoperative tests [32]. Moreover, these tests (if done routinely) have the potential to cause harm and incur unnecessary costs without significantly altering preoperative management [33]. Cardiovascular instability monitoring was the second most frequently administered preoperative treatment, given to 13 patients or 20% of the sample. The need for monitoring cardiovascular stability is subjective, and our study considered patients with specific cardiovascular conditions and end-organ failure that may necessitate monitoring (S1 Table). Traditional hospital admission is more suitable for these conditions according to some guidelines [13,34,35] and accounted for 3.2% of our study population (13 patients).

The strength of this study lies in its relatively complete data set and adequate follow-up. Only five patients (1.2%) were lost to follow-up and could not be reached by phone. This means the study findings reflect real-world practice data. The traditional approach to preoperative admission is still prevalent in many countries, including those in Asia and other developing nations [6–8]. This study's representation of actual practice data provides valuable insights into overnight-stay LC. The study findings have had a significant impact on the implementation of a daycare surgery protocol for LC at the study hospital. One effective approach involved carefully selecting patients to eliminate the need for preoperative admission and instead implementing an efficient PAC. As a result, over the past year, the hospital has successfully performed approximately 100 daycare LC procedures. These achievements have been recognized at the national level; the hospital received the Best Practice Award from the Thailand One day surgery and Minimally Invasive Surgery national forum in 2022.

This study had several limitations to consider. First, whereas our findings suggest that preoperative admission could potentially be omitted, the study was not designed to test the safety of this omission in terms of morbidity or mortality. Whereas previous studies have shown the safety of surgery without hospital admission [36,37], those results can be used to provide context for our results. Second, our results regarding treatment requirements were based on somewhat subjective criteria. There are varying approaches to treatment criteria for different conditions (e.g., hypokalemia, hyponatremia, blood sugar control). For example, management of anticoagulant bridging or switching can be performed on an outpatient or inpatient basis [38], and in our study, anticoagulant bridging was considered a mandatory inpatient treatment. Hypokalemia, which was the most prevalent case of over-treatments in our data, also

varied in terms of the preoperative correction threshold [39], even among the anesthesiologists at our hospital. It is important to keep these variations in mind when interpreting our results, which are presented in S1 Table. In connection with the second limitation, two participating specialists who assessed the primary outcome were aware of the study's hypotheses. Consequently, there is a possibility of bias in their assessments of whether preoperative treatment was necessary. Third, owing to the low incidence of medical complications in this study, the sample size may not be sufficient to capture significant variables in the multivariable analysis (Table 3). Finally, the lack of data on obstructive sleep apnea (OSA), an important factor to consider in daycare surgery [34], was another limitation of this study. Although OSA was screened during a pre-anesthesia consultation, our hospital's records were inconsistent, and it was not included in the analysis.

## Conclusions

Our study suggests that preoperative admission for elective LC does not offer substantial therapeutic benefits for most patients. Although traditional preoperative admission is safe, with low rates of morbidity, it should be reassessed in terms of optimizing resource utilization. Our findings indicate that patients' comorbidities have a greater impact on preoperative management than age, obesity, and ASA status.

## Supporting information

**S1 Table. Criteria for evaluating the need for preoperative treatment in the study.**
(DOCX)

**S2 Table. Sensitivity analysis results for patients with symptomatic gallstones only (excluding other indications for laparoscopic cholecystectomy, n = 295) and for elective laparoscopic cholecystectomy without additional procedures (excluding patients who underwent intraoperative cholangiography, n = 377).**
(DOCX)

**S1 File. STROBE statement—checklist of items that should be included in reports of *cohort studies* Study's title: Preoperative admission is non-essential in most patients receiving elective laparoscopic cholecystectomy: A cohort study.**
(PDF)

## Acknowledgments

We express our gratitude to Dr. Sanguansak Siangruangsang for providing valuable suggestions regarding the preoperative treatment criteria. Our appreciation also goes to Dr. Wibun Phanthabordeekorn for his support in starting the ambulatory surgery program in our hospital. We thank Analisa Avila, MPH, ELS, of Edanz (www.edanz.com/ac) for editing a draft of this manuscript.

## Author Contributions

**Conceptualization:** Suppadech Tunruttanakul, Ratchanee Tunruttanakul, Kamoltip Prasopsuk, Kwanhathai Sakulsansern, Kyrhatii Trikhirhisthit.

**Data curation:** Suppadech Tunruttanakul, Kamoltip Prasopsuk.

**Formal analysis:** Suppadech Tunruttanakul, Kamoltip Prasopsuk, Kwanhathai Sakulsansern, Kyrhatii Trikhirhisthit.

**Funding acquisition:** Suppadech Tunruttanakul.

**Investigation:** Suppadech Tunruttanakul, Ratchanee Tunruttanakul, Kamoltip Prasopsuk, Kwanhathai Sakulsansern, Kyrhatii Trikhirhisthit.

**Methodology:** Suppadech Tunruttanakul, Ratchanee Tunruttanakul, Kamoltip Prasopsuk, Kwanhathai Sakulsansern, Kyrhatii Trikhirhisthit.

**Software:** Suppadech Tunruttanakul, Kyrhatii Trikhirhisthit.

**Validation:** Suppadech Tunruttanakul, Ratchanee Tunruttanakul, Kamoltip Prasopsuk, Kwanhathai Sakulsansern, Kyrhatii Trikhirhisthit.

**Visualization:** Suppadech Tunruttanakul.

**Writing – original draft:** Suppadech Tunruttanakul, Ratchanee Tunruttanakul, Kamoltip Prasopsuk, Kwanhathai Sakulsansern, Kyrhatii Trikhirhisthit.

**Writing – review & editing:** Suppadech Tunruttanakul, Ratchanee Tunruttanakul, Kamoltip Prasopsuk, Kwanhathai Sakulsansern, Kyrhatii Trikhirhisthit.

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
