## [Decision Letter · Decision Letter 0]

28 Jun 2023

PONE-D-23-13647Preoperative admission is non-essential in most patients receiving elective laparoscopic cholecystectomy: a cohort study

PLOS ONE

Dear Dr. Tunruttanakul,

Thank you for submitting your manuscript to PLOS ONE. After careful consideration, we feel that it has merit but does not fully meet PLOS ONE’s publication criteria as it currently stands. Therefore, we invite you to submit a revised version of the manuscript that addresses the points raised during the review process.

The study is interesting when contextualized in the geographic area where the hospital is located. I believe that the reviewers and in particular reviewer 3 have focused on the changes that the paper needs in order to meet the criteria for publication.

Certainly a considerable amount of redrafting effort is required by the authors. To the reviewers’ considerations i add my own.

Because the study is focused on patients treated with laparoscopic cholecystectomy in elective setting it is necessary to clarify the pathway of patients with acute cholecystitis and biliary pancreatitis . In fact in the materials and methods it is stated: : “We included only patients with gallstone pancreatitis or acute cholecystitis who showed improvement or resolution of symptoms after initial conservative treatment “. Improvement of acute cholecystitis or pancreatitis does not mean cure for this reason the surgery cannot be considered elective.

What definition or guideline is being referred to in order to state that “The patient's age was recorded in years and grouped into two categories: below 65 years old and 65 years old or older (according to the definition of elderly)”

It is necessary to clarify at what stage the "preoperative anesthesiological examinations (PAC)" are performed.

How is the PAC organized? Are there any elective patients who enter the operating room without prior sighting by the anesthesiologist? Is there a check list in your hospital to get access to the operating theatre?

Patients who have choledocholithiasis should be investigated apart because they represent a population with a different disease than symptomatic gallstones.

What are the criteria to perform “intraoperative cholangiography, endoscopic retrograde cholangiography (ERC), and laparoscopic bile duct exploration”.

What are the criteria that must be met for discharge?

“Finally, this study lacked information on canceled surgeries”. If patients were not operated on because of the finding of problems during the preoperative evaluation by the anesthesiologist or by the ward surgeon the absence of these patients could completely change the value of the paper. This is a crucial point that needs to be explained in some way.

Since the authors are familiar with the national scenario, do they think it is possible to suggest the pathway to be organized for this sort of patient?

We look forward to receiving your revised manuscript.

Kind regards,

Fabrizio D'Acapito, Ph.D,M.D.

Academic Editor

PLOS ONE

“This study received financial support from Sawanpracharak Hospital Medical Education center (https://mec.spr.go.th).”

4. We note that Figure 2 in your submission contain copyrighted images. All PLOS content is published under the Creative Commons Attribution License (CC BY 4.0), which means that the manuscript, images, and Supporting Information files will be freely available online, and any third party is permitted to access, download, copy, distribute, and use these materials in any way, even commercially, with proper attribution. For more information, see our copyright guidelines: http://journals.plos.org/plosone/s/licenses-and-copyright.

b.If you are unable to obtain permission from the original copyright holder to publish these figures under the CC BY 4.0 license or if the copyright holder’s requirements are incompatible with the CC BY 4.0 license, please either i) remove the figure or ii) supply a replacement figure that complies with the CC BY 4.0 license. Please check copyright information on all replacement figures and update the figure caption with source information. If applicable, please specify in the figure caption text when a figure is similar but not identical to the original image and is therefore for illustrative purposes only.

Additional Editor Comments:

The study is interesting when contextualized in the geographic area where the hospital is located. I believe that the reviewers and in particular reviewer 3 have focused on the changes that the paper needs in order to meet the criteria for publication.

Certainly a considerable amount of redrafting effort is required by the authors. To the reviewers’ considerations i add my own.

Because the study is focused on patients treated with laparoscopic cholecystectomy in elective setting it is necessary to clarify the pathway of patients with acute cholecystitis and biliary pancreatitis . In fact in the materials and methods it is stated: : “We included only patients with gallstone pancreatitis or acute cholecystitis who showed improvement or resolution of symptoms after initial conservative treatment “. Improvement of acute cholecystitis or pancreatitis does not mean cure for this reason the surgery cannot be considered elective.

What definition or guideline is being referred to in order to state that “The patient's age was recorded in years and grouped into two categories: below 65 years old and 65 years old or older (according to the definition of elderly)”

It is necessary to clarify at what stage the "preoperative anesthesiological examinations (PAC)" are performed.

How is the PAC organized? Are there any elective patients who enter the operating room without prior sighting by the anesthesiologist? Is there a check list in your hospital to get access to the operating theatre?

Patients who have choledocholithiasis should be investigated apart because they represent a population with a different disease than symptomatic gallstones.

What are the criteria to perform “intraoperative cholangiography, endoscopic retrograde cholangiography (ERC), and laparoscopic bile duct exploration”.

What are the criteria that must be met for discharge?

“Finally, this study lacked information on canceled surgeries”. If patients were not operated on because of the finding of problems during the preoperative evaluation by the anesthesiologist or by the ward surgeon the absence of these patients could completely change the value of the paper. This is a crucial point that needs to be explained in some way.

Since the authors are familiar with the national scenario, do they think it is possible to suggest the pathway to be organized for this sort of patient?

Reviewers' comments:

Reviewer's Responses to Questions

**Comments to the Author**

1. Is the manuscript technically sound, and do the data support the conclusions?

Reviewer #1: Yes

Reviewer #2: Yes

Reviewer #3: No

2. Has the statistical analysis been performed appropriately and rigorously? 

Reviewer #1: Yes

Reviewer #2: Yes

Reviewer #3: No

3. Have the authors made all data underlying the findings in their manuscript fully available?

Reviewer #1: Yes

Reviewer #2: Yes

Reviewer #3: No

4. Is the manuscript presented in an intelligible fashion and written in standard English?

Reviewer #1: Yes

Reviewer #2: Yes

Reviewer #3: No

5. Review Comments to the Author

Reviewer #1: Thank you for allowing me to review your manuscript "Preoperative admission is non-essential in most patients receiving elective laparoscopic cholecystectomy: a cohort study" for PLoS ONE. I enjoyed reading it and its premise that pre-operative assessment for patients undergoing laparoscopic cholecystectomy has a poor yield and could be safely omitted in the majority of patients.

A couple of comments which I hope will strengthen your manuscript.

Firstly, your study currently included 50 patients who had choledocholithiasis requiring intraoperative or postoperative bile duct clearance. I believe these patients should be excluded so that you analyze and present a homogenous treatment group (those who underwent elective laparoscopic cholecystectomy alone), which would strengthen your observations and reccomendation.

Second, your study includes 67 patients classed as ASA I; none of these were subsequently reported to have undergone pre-admission assessment or treatment. Some of these patients may overlap with those with CBD stones in my point above. As it would not be expected that these ASA I patients undergo pre-operative testing (as per guidelines), I would exclude these patients from your analysis.

In so doing, your hypothesis would be clearer -- that pre-admission assessment and treatment of patients with medical comorbidities (ASA II and III) (advocated by some medical professional societies), does not add value to the patient's care. It would also make any recommendations based on interpretation of your results stronger.

Reviewer #2: In the article in question the different preoperative treatments for the different pathologies for which patients undergo surgery are not well differentiated, i.e. they are grouped into a single group: cholelithiasis, biliary pancreatitis, acute cholecystitis and choledocholithiasis. I think it is important to differentiate between the different pathologies which certainly benefit from different preoperative treatments. I would also like to understand if these patients underwent outpatient preoperative tests (blood tests, electrocardiogram, chest X-ray, anaesthesiology visit), or all of this is performed the day before or the day of the operation. These exams would certainly help to differentiate the different ASAs and consequently the preoperative preparation.

The fact of correcting the potassium seems a confusing data, as it resulted in the statistical analysis as an over-treatment.

The sample size is rather small, but it is not clear how many beds the hospitals in which the patients were hospitalized and operated on have.

Reviewer #3: 1. This is a retrospective cohort study looking at the value of preoperative admission for laparoscopic cholecystectomy. The authors concluded that 87% of preoperative admissions provided no benefit and were therefore unnecessary. It is not clear what this is based on. Are the authors saying that 87% of patients had no procedures or treatments done during their pre-surgery day in the hospital that needed to be done on that day, or that needed to be done while they were admitted. In other words, only 13% of the patients had procedures done that needed to be done the day before surgery during a pre-surgery admission day. Please state this more clearly.

2. In the Introduction, the authors are trying to explain what is being reviewed by the 2 raters, but this section is unclear. The category they call ‘documented treatment provided’ sounds like it should be called ‘necessary treatment’, meaning that the procedure was needed, based on the patient’s presenting condition. It should be stated here that each treatment is being rated by the 2 raters, so the unit of analysis is treatments, not patients. However, on P 14 it appears that patients is the unit of analysis, when the authors discuss having 75 patients of whom 38 had potassium correction. Please make it more clear how the outcome variables were calculated. Since some treatments a patient received may have been necessary, while others may have been unnecessary, I’m not sure how patients could be used as the unit of analysis, since this would require that each patient be labeled as either receiving necessary treatment during their pre-operative day in the hospital, or not receiving necessary treatment. What about patients who received both?

3. Please re-write this paragraph (it is extremely confusing): “ The final categorization of preoperative treatment was summarized into documented treatment provided and required treatment. Required treatment refers to documented treatment provided, excluding overtreatment and including under-treatment. Treatment that could be given as an outpatient visit was required treatment that could be provided as an outpatient. All necessary preoperative cardiovascular monitoring was also considered preoperative treatment (S1 Table). “

4. This is unclear: “The secondary outcomes of the study were postoperative morbidity, mortality within 30 days, and length of hospital stay (LOHS).”

5. Length of hospital stay is likely to be very highly positively skewed, so it can’t be used as the dependent variable in multiple regression unless it is transformed to make it more normally distributed. The authors should try a log10 transformation, but this may still not correct all the outliers. If there are a few very long admissions, these few patients will have too large an influence on the results. One option might be to recode length of stay into deciles (coded 1-10) to use in linear regression, or consider time to event analysis (where discharge is the event), or consider Poisson regression (which has many complications), or possibly dichotomize this and use logistic regression.

6. Given that providers are accustomed to having a day of pre-surgery admission, it is likely that outcomes will not be as good as the authors expect if there is a sudden switch and they eliminate this pre-surgery admission. Hospital processes will need to be established based on same-day surgery that do not currently exist.

6. PLOS authors have the option to publish the peer review history of their article (what does this mean?). If published, this will include your full peer review and any attached files.

Reviewer #1: No

Reviewer #2: No

Reviewer #3: No

---

## [Author Response · Author response to Decision Letter 0]

1 Aug 2023

Responses to the Additional Editor Comments

We express our sincere gratitude to Dr. Fabrizio D'Acapito for providing us with detailed and valuable comments. Your valuable input has been instrumental in improving our research. We have taken each comment into careful consideration and made the necessary revisions accordingly. Here is a summary of how we addressed each comment:

Comment 1. The study is interesting when contextualized in the geographic area where the hospital is located.

 Response: We sincerely appreciate Dr. D'Acapito for your kind and encouraging comment.

Comment 2. Because the study is focused on patients treated with laparoscopic cholecystectomy in elective setting it is necessary to clarify the pathway of patients with acute cholecystitis and biliary pancreatitis. In fact in the materials and methods it is stated: “We included only patients with gallstone pancreatitis or acute cholecystitis who showed improvement or resolution of symptoms after initial conservative treatment “. Improvement of acute cholecystitis or pancreatitis does not mean cure for this reason the surgery cannot be considered elective.

 Response: We extend our gratitude to Associate Professor D'Acapito for bringing up this crucial issue. We sincerely apologize for our oversight and appreciate your valuable input. We have carefully addressed the concern by deleting and rephrasing certain sentences to align with the corrected information.

 (From line 7, p 6 in the Material and Methods)

 We deleted sentences “LC indications included symptomatic gallstones, gallstone pancreatitis, acute cholecystitis, and choledocholithiasis. We included only patients with gallstone pancreatitis or acute cholecystitis who showed improvement or resolution of symptoms after initial conservative treatment and subsequently underwent elective LC.”

 We revised and added sentences “We included various gallstone-related indications for elective LC, encompassing symptomatic gallstones, acute cholecystitis, gallstone pancreatitis, and choledocholithiasis. For cases of acute cholecystitis or gallstone pancreatitis, we only included patients who showed symptom improvement or resolution after initial conservative treatment and who subsequently underwent elective LC. Patients who underwent early LC during the first episode of acute cholecystitis or gallstone pancreatitis, which were considered emergency cases, were excluded from the study. Additionally, patients with recurrent biliary admissions for either condition that necessitated urgent LC before the scheduled procedure were also excluded” 

Comment 3. What definition or guideline is being referred to in order to state that “The patient's age was recorded in years and grouped into two categories: below 65 years old and 65 years old or older (according to the definition of elderly)”

 Response: We thank Dr. D'Acapito for providing this insightful comment. We sincerely apologize once again for our oversight in lacking the necessary information. In response to this issue, we have promptly addressed it by deleting and rephrasing the relevant sentence accordingly.

 (From line 26, p 5 in the Material and Methods)

 We have deleted sentences “according to the definition of elderly” and replaced it with “according to the most common definition used in medical research” with added reference No. 13 

Comment 4&5. It is necessary to clarify at what stage the "preoperative anesthesiological examinations (PAC)" are performed.

 And

 How is the PAC organized? Are there any elective patients who enter the operating room without prior sighting by the anesthesiologist? Is there a check list in your hospital to get access to the operating theatre?

 Response: We sincerely appreciate Dr. D'Acapito's valuable comments, which have greatly contributed to the improvement of our manuscript. We are grateful for bringing these important issues to our attention. In response to these comments, we have made the necessary revisions to address them. Specifically, we have added detailed information about our Preoperative Anesthesiology Clinic (PAC) system in the Material and Methods section as follows:

 (From line 11, p 7 in the Material and Methods)

 We have included a new paragraph 

 “PAC visits were conducted for all patients undergoing elective surgeries at our hospital as outpatient consultations and took place on the same day if surgeons and patients agreed to proceed with planned surgeries. Anesthesiologists, along with nurse anesthetists, conducted all pre-anesthetic evaluations. Additional consultations with subspecialists could be arranged if the anesthesiologist required more specific evaluations. Occasionally, the absence of PAC visits in our hospital could occur owing to human error, either on the part of patients or the administrative staff. To ensure comprehensive assessments, in-patient preoperative anesthesiology visits and checklists were routinely completed on the day of preoperative admission. This allowed for further patient evaluation and the execution of necessary treatments. In cases where patients missed PAC visits, they would still undergo in-patient preoperative anesthesiology visits to prevent cancellations. However, surgical cancellations could still occur if patients were deemed unfit for the planned operations. All surgeries had to be approved by the anesthesiologists before the operation could be performed. Preoperative screening tests were conducted during both the PAC visits and in-patient preoperative anesthesiology visits. Complete blood count and chest radiography were performed for all patients. Patients over 45 years old underwent electrocardiogram testing, and those over 60 years old had blood urea nitrogen, creatinine, and electrolyte levels further assessed. Additional laboratory tests were organized based on specific comorbidities, such as blood sugar testing for diabetes mellitus and liver function tests for suspected liver diseases.”

Comment 6. Patients who have choledocholithiasis should be investigated apart because they represent a population with a different disease than symptomatic gallstones.

 Response: We greatly appreciate Dr. D'Acapito's insightful comment. As an elective laparoscopic cholecystectomy is commonly performed for patients with choledocholithiasis, it is true that some of the patients included in our study fell into this category. We acknowledge the importance of this consideration. In order to address this concern while ensuring the study's applicability across various clinical scenarios, we excluded patients with choledocholithiasis who required additional complex procedures such as laparoscopic bile duct exploration or pre-operative same-admission endoscopic retrograde cholangiography. However, we retained patients with choledocholithiasis who underwent investigation and treatment through endoscopic retrograde cholangiography in separate admission sessions, or those who underwent only the laparoscopic cholecystectomy procedure during the study period. Furthermore, we included patients who underwent intraoperative cholangiography during the laparoscopic cholecystectomy, as they had a low risk of choledocholithiasis and the additional procedure had minimal impact in terms of time and invasiveness. With these exclusions, we believe that our study's objective will be more clearly defined while still maintaining its broad applicability. Lastly, we conducted a sensitivity analysis to examine a more homogeneous patient population. This analysis involved excluding all other laparoscopic cholecystectomy indications and removing intraoperative cholangiography from the analysis. We have made numerous revisions in line with this exclusion, as outlined below.

 1.) We excluded patients who had undergone more complex procedures, but we retained the practice of performing intraoperative cholangiography. The message has been added as follows:

 (From line 5, p 5 in the Material and Methods)

 “Because our focus was on patients undergoing elective LC, we excluded cases involving more complex adjunctive procedures such as laparoscopic bile duct exploration and same-admission preoperative endoscopic retrograde cholangiography. However, we included patients receiving elective LC who underwent intraoperative cholangiography (IOC) because it is a relatively simple procedure with minimal additional time requirements [11]. Furthermore, many countries have adopted a routine approach to IOC for all patients undergoing LC [12].” We have added references 11 and 12 accordingly.

 With more exclusions, the study population decreased from 427 to 405, resulting in changes in numbers throughout the manuscript. We have indicated these numbers changes by using strikethrough font [example] to indicate deleted numbers and highlighting revised numbers in red highlight.

 2.) We have revised and deleted sentences related to choledocholithiasis as follows:

 (From line 10, p 6 in the Material and Methods)

 The message “We included various gallstone-related indications for elective LC, encompassing symptomatic gallstones, acute cholecystitis, gallstone pancreatitis, and choledocholithiasis.” has been added. 

 (From line 18, p 6 in the Material and Methods)

 The message “Choledocholithiasis included only patients who had previously undergone endoscopic retrograde cholangiography to remove their stones and were scheduled for elective LC.” has been added. 

 (From line 23, p 6 in the Material and Methods)

 We deleted sentences regarding additional procedures as follow “Additional procedures performed in the hospital were recorded and commonly included intraoperative cholangiography, endoscopic retrograde cholangiography (ERC), and laparoscopic bile duct exploration. Only ERC performed during the same admission as LC was recorded as a supplemental procedure; ERC completed in different hospital admission episodes was not registered as the presence of a supplemental procedure. The other two procedures were always performed with LC.” 

 (From line 2, p 11 in the Results)

 The statements regarding further exclusion was revised or added as follow: ”The records of 62 patients were excluded owing to being under the age of 18 years, having undergone emergency surgery, and having laparoscopic bile duct exploration or same-admission preoperative endoscopic retrograde cholangiography. There was no data loss. Overall, a total of 405 patients were included in the analysis.” and the statements regarding previous exclusion criteria was deleted as follow: “The records of 39 patients were excluded owing to being under the age of 18 years or having undergone emergency surgery, with no data loss.” (From line 26, p 10 in the Results)

 3.) We added sentences expressing reason for maintaining some patients’ populations as follow: 

 (From line 20, p 6 in the Material and Methods) 

 “We maintained data heterogeneity to address the diverse indications commonly seen in elective LC, ensuring the relevance and applicability of our study.”

 4.) We added sentences regarding sensitivity analysis as follow:

 (From line 17, p 10 in the Material and Methods)

 “Finally, sensitivity analysis was conducted to investigate outcome variability according to alteration of LC indications and the presence of additional IOC.”

 (From line 12, p 15 in the Results)

 We added a paragraph regarding the sensitivity analysis result: “Sensitivity analyses were conducted to explore the impact of more homogeneous data on the main outcome (required treatment); the results are summarized in S2 Table. When only patients with symptomatic gallstones were included (excluding other indications for LC, n=295), and when only elective LC without additional procedures was examined (excluding patients who underwent IOC, n=377), the findings were consistent with the full model. In both sensitivity analyses, the presence of comorbidities was the only significant factor associated with the need for preoperative treatment. The odds of requiring preoperative treatment increased with the number of comorbidities: odds ratios (95% confidence intervals) were 11.0 (2.2, 54.1), 35.2 (6.4, 192.6), and 110.0 (13.5, 896.9) for one, two, and three comorbidities, respectively, in the model with symptomatic gallstones as the sole indication. Similarly, in the model without additional IOC, the odds ratios (95% confidence intervals) were 8.6 (2.3, 32.6), 28.1 (6.8, 116.4), and 124.2 (18.9, 814.6) for one, two, and three comorbidities, respectively. Other factors were not found to be associated with preoperative treatment in either of the sensitivity analytic models.” 

 5.) We added S2 Table to detail the sensitivity analysis result. 

Comment 7. What are the criteria to perform “intraoperative cholangiography, endoscopic retrograde cholangiography (ERC), and laparoscopic bile duct exploration”.

 And 

 What are the criteria that must be met for discharge?

 Response: We sincerely appreciate Dr. D'Acapito for providing these valuable suggestions. We would like to address both of these comments collectively. Regarding the exclusion mentioned earlier, we have omitted patient data related to endoscopic retrograde cholangiography (ERC) and laparoscopic bile duct exploration from our manuscript. As a result, criteria for these procedures are no longer applicable. However, we have included specific criteria for intraoperative cholangiography and for discharge as follows:

 (From line 1, p 7 in the Material and Methods)

 We incorporated criteria for performing intraoperative cholangiography (IOC) as follows: “Regarding the presence of IOC parameters, in our department, selective IOC was indicated for patients with symptomatic gallstones who had a relatively low but non-negligible risk of choledocholithiasis. This included patients with a clinical history of jaundice, pancreatitis, elevated bilirubin levels, and abnormal liver function test results. In cases with a greater suspicion of choledocholithiasis, alternative modalities such as magnetic resonance cholangiopancreatography or endoscopic retrograde cholangiography were used [16]. Choledocholithiasis detected by IOC was managed based on the decision of the attending surgeon. Treatment options included transcystic bile duct stent placement [17] or performing endoscopic retrograde cholangiography during the same procedure [18].” We have added references 16, 17, and 18 accordingly.

 (From line 16, p 9 in the Material and Methods)

 To address the discharge criteria, we have included the following statement: “Discharge considerations for patients were determined by the attending surgeon. In general, the criteria for discharge from our department included stable vital signs, normal breathing with oxygen saturation levels above 90% on room air, reasonable mobility, and a manageable level of pain, nausea, and vomiting.”

Comment 8. “Finally, this study lacked information on canceled surgeries”. If patients were not operated on because of the finding of problems during the preoperative evaluation by the anesthesiologist or by the ward surgeon the absence of these patients could completely change the value of the paper. This is a crucial point that needs to be explained in some way.

 Response: We extend our heartfelt gratitude to Dr. D'Acapito for raising this critical point in our study. We sincerely apologize for our previous oversight in not conducting a thorough search for surgery cancellation data. However, we have made concerted efforts to retrieve the necessary information on surgery cancellations. As a result, we have successfully obtained the data and incorporated it into our study. We have included relevant sentences addressing this comment in multiple sections of the manuscript.

 (From line 21, p 5 in the Material and Methods)

 We added surgery cancellations to the reviewing parameter. The full sentence was “In this study, we reviewed the following parameters that could potentially impact the study objective: patient's age, sex, body mass index (BMI), American Society of Anesthesiologists (ASA) status, comorbidities, presence of preoperative anesthesiology clinic (PAC) visits, and LC indications, presence of additional IOC, and surgery cancellations.”

 (From line 8, p 9 in the Material and Methods)

 We added more sentences regarding surgery cancellations in the paragraph regarding the preoperative treatment requirement evaluation as follow: “Only surgery cancellations owing to medical reasons [19] were considered required treatment. The same criteria used to evaluate preoperative treatment were applied to patients with other reasons for cancellation.” We have also added reference 19. 

 (From line 29, p 11 in the Results)

 We added sentences regarding surgery cancellations in the paragraph concerning the Table 1 as follow “Surgery cancellations related to the study protocol were observed in two cases (0.5%) out of a total of 798 cancellations during the study period. One cancellation was for a medical reason (recently treated hyperthyroidism), and the other was attributed to the unavailability of theater time. The patient who canceled surgery for medical reasons later refused to undergo the procedure.”

 We added the results relating to surgery cancellations Table 1 and Table 2 shown in red highlight. 

 As one medical cancellation case refused to undergo further surgery resulting low patient number for analysis of medical complication and hospital stay. We addressed this by adding sentence “The patient whose surgery was canceled refused to undergo the procedure, resulting in a decrease of one in the number of patients analyzed for medical complications and prolonged LOHS (n=404 instead of the total n=405).” on line 11 page 14 in the Results and added footnote “n=405 for required treatment outcome, n=404 for medical complication outcome, and n=404 for prolonged hospital stay outcome. One patient who had a surgery cancellation refused to undergo surgery and was excluded from the analysis.” In Table 3. 

Comment 9. Since the authors are familiar with the national scenario, do they think it is possible to suggest the pathway to be organized for this sort of patient?

Response: We appreciate Dr. D'Acapito for this valuable suggestion. With the findings of our study, we have realized that preoperative admission was not necessary for the majority of laparoscopic cholecystectomy patients. In response, we have revised our approach and implemented an efficient Preoperative Anesthesiology Clinic (PAC) system. This change in practice has garnered recognition at the national level. In line with these developments, we have added the following sentences to the Discussion section (From line 8, p 20): “The study findings have had a significant impact on the implementation of a daycare surgery protocol for LC at the study hospital. One effective approach involved carefully selecting patients to eliminate the need for preoperative admission and instead implementing an efficient PAC. As a result, over the past year, the hospital has successfully performed approximately 100 daycare LC procedures. These achievements have been recognized at the national level; the hospital received the Best Practice Award from the Thailand One-day surgery and Minimally Invasive Surgery national forum in 2022.” 

Responses to the comments of Reviewer #1

We are grateful to Reviewer #1 for their insightful comments, which have played a crucial role in elevating the quality of our research. We carefully considered each of your suggestions, and the following are our responses to address your valuable feedback:

Comment 1 Thank you for allowing me to review your manuscript "Preoperative admission is non-essential in most patients receiving elective laparoscopic cholecystectomy: a cohort study" for PLoS ONE. I enjoyed reading it and its premise that pre-operative assessment for patients undergoing laparoscopic cholecystectomy has a poor yield and could be safely omitted in the majority of patients.

Response: We extend our sincere gratitude to Reviewer #1 for providing this encouraging comment. Your positive feedback is greatly appreciated.

A couple of comments which I hope will strengthen your manuscript.

Comment 2 Firstly, your study currently included 50 patients who had choledocholithiasis requiring intraoperative or postoperative bile duct clearance. I believe these patients should be excluded so that you analyze and present a homogenous treatment group (those who underwent elective laparoscopic cholecystectomy alone), which would strengthen your observations and recommendation.

Response: We express our gratitude to Reviewer #1 for bringing this important issue to our attention, and we wholeheartedly agree with the comment provided. As it addresses a similar concern raised by the editor, we aim to maintain consistency in our responses. Consequently, we are providing a consistent response to both the reviewer and the editor. As an elective laparoscopic cholecystectomy is commonly performed for patients with choledocholithiasis, it is true that some of the patients included in our study fell into this category. We acknowledge the importance of this consideration. In order to address this concern while ensuring the study's applicability across various clinical scenarios, we excluded patients with choledocholithiasis who required additional complex procedures such as laparoscopic bile duct exploration or pre-operative same-admission endoscopic retrograde cholangiography. However, we retained patients with choledocholithiasis who underwent investigation and treatment through endoscopic retrograde cholangiography in separate admission sessions, or those who underwent only the laparoscopic cholecystectomy procedure during the study period. Furthermore, we included patients who underwent intraoperative cholangiography during the laparoscopic cholecystectomy, as they had a low risk of choledocholithiasis and the additional procedure had minimal impact in terms of time and invasiveness. With these exclusions, we believe that our study's objective will be more clearly defined while still maintaining its broad applicability. Lastly, we conducted a sensitivity analysis to examine a more homogeneous patient population. This analysis involved excluding all other laparoscopic cholecystectomy indications and removing intraoperative cholangiography from the analysis. We have made numerous revisions in line with this exclusion, as outlined below.

 1.) We excluded patients who had undergone more complex procedures, but we retained the practice of performing intraoperative cholangiography. The message has been added as follows:

 (From line 5, p 5 in the Material and Methods)

 “Because our focus was on patients undergoing elective LC, we excluded cases involving more complex adjunctive procedures such as laparoscopic bile duct exploration and same-admission preoperative endoscopic retrograde cholangiography. However, we included patients receiving elective LC who underwent intraoperative cholangiography (IOC) because it is a relatively simple procedure with minimal additional time requirements [11]. Furthermore, many countries have adopted a routine approach to IOC for all patients undergoing LC [12].” We have added references 11 and 12 accordingly.

 With more exclusions, the study population decreased from 427 to 405, resulting in changes in numbers throughout the manuscript. We have indicated these numbers changes by using strikethrough font [example] to indicate deleted numbers and highlighting revised numbers in red highlight. 

 2.) We have revised and deleted sentences related to choledocholithiasis as follows:

 (From line 10, p 6 in the Material and Methods)

 The message “We included various gallstone-related indications for elective LC, encompassing symptomatic gallstones, acute cholecystitis, gallstone pancreatitis, and choledocholithiasis.” has been added. 

 (From line 18, p 6 in the Material and Methods)

 The message “Choledocholithiasis included only patients who had previously undergone endoscopic retrograde cholangiography to remove their stones and were scheduled for elective LC.” has been added. 

 (From line 23, p 6 in the Material and Methods)

 We deleted sentences regarding additional procedures as follow “Additional procedures performed in the hospital were recorded and commonly included intraoperative cholangiography, endoscopic retrograde cholangiography (ERC), and laparoscopic bile duct exploration. Only ERC performed during the same admission as LC was recorded as a supplemental procedure; ERC completed in different hospital admission episodes was not registered as the presence of a supplemental procedure. The other two procedures were always performed with LC.” 

 (From line 2, p 11 in the Results)

 The statements regarding further exclusion was revised or added as follow: ”The records of 62 patients were excluded owing to being under the age of 18 years, having undergone emergency surgery, and having laparoscopic bile duct exploration or same-admission preoperative endoscopic retrograde cholangiography. There was no data loss. Overall, a total of 405 patients were included in the analysis.” and the statements regarding previous exclusion criteria was deleted as follow: “The records of 39 patients were excluded owing to being under the age of 18 years or having undergone emergency surgery, with no data loss.” (From line 26, p 10 in the Results)

 3.) We added sentences expressing reason for maintaining some patients’ populations as follow: 

 (From line 20, p 6 in the Material and Methods) 

 “We maintained data heterogeneity to address the diverse indications commonly seen in elective LC, ensuring the relevance and applicability of our study.”

 4.) We added sentences regarding sensitivity analysis as follow:

 (From line 17, p 10 in the Material and Methods)

 “Finally, sensitivity analysis was conducted to investigate outcome variability according to alteration of LC indications and the presence of additional IOC.”

 (From line 12, p 15 in the Results)

 We added a paragraph regarding the sensitivity analysis result: “Sensitivity analyses were conducted to explore the impact of more homogeneous data on the main outcome (required treatment); the results are summarized in S2 Table. When only patients with symptomatic gallstones were included (excluding other indications for LC, n=295), and when only elective LC without additional procedures was examined (excluding patients who underwent IOC, n=377), the findings were consistent with the full model. In both sensitivity analyses, the presence of comorbidities was the only significant factor associated with the need for preoperative treatment. The odds of requiring preoperative treatment increased with the number of comorbidities: odds ratios (95% confidence intervals) were 11.0 (2.2, 54.1), 35.2 (6.4, 192.6), and 110.0 (13.5, 896.9) for one, two, and three comorbidities, respectively, in the model with symptomatic gallstones as the sole indication. Similarly, in the model without additional IOC, the odds ratios (95% confidence intervals) were 8.6 (2.3, 32.6), 28.1 (6.8, 116.4), and 124.2 (18.9, 814.6) for one, two, and three comorbidities, respectively. Other factors were not found to be associated with preoperative treatment in either of the sensitivity analytic models.” 

 5.) We added S2 Table to detail the sensitivity analysis result.

Comment 3 Second, your study includes 67 patients classed as ASA I; none of these were subsequently reported to have undergone pre-admission assessment or treatment. Some of these patients may overlap with those with CBD stones in my point above. As it would not be expected that these ASA I patients undergo pre-operative testing (as per guidelines), I would exclude these patients from your analysis.

In so doing, your hypothesis would be clearer -- that pre-admission assessment and treatment of patients with medical comorbidities (ASA II and III) (advocated by some medical professional societies), does not add value to the patient's care. It would also make any recommendations based on interpretation of your results stronger.

Response: We sincerely appreciate Reviewer #1 for providing this insightful comment. Your observation is indeed accurate, and we wholeheartedly agree that pre-operative testing for ASA I patients is futile. We understand that this practice might be common in our country, but we acknowledge the importance of raising awareness about its inefficiency.

 After carefully considering your concerns, we have decided to retain ASA I patients in our data to highlight the wastefulness of the current practice. We greatly value your input, and as a result, we have added sentences in the Results section (From line 12, p 13) stating: “Notably, none of the patients with ASA status I received both actual treatment rendered and the required treatment.” Additionally, in the Discussion section (From line 12, p 18), we have included the following statement: “Patients with lower ASA status demonstrated a lower rate of preoperative treatment, with no patients who had ASA status I requiring treatment, and only 8.9% of those with ASA status II receiving the required treatment.”

 We hope that our study's results will bring attention to the unnecessary pre-operative testing for ASA I patients and prompt a change in this practice. 

Responses to the comments of Reviewer #2

Our sincere thanks go to Reviewer #2 for their valuable and constructive comments, which have greatly contributed to the improvement of our study. We carefully considered each of your remarks and have provided the following responses:

Comment 1 In the article in question the different preoperative treatments for the different pathologies for which patients undergo surgery are not well differentiated, i.e. they are grouped into a single group: cholelithiasis, biliary pancreatitis, acute cholecystitis and choledocholithiasis. I think it is important to differentiate between the different pathologies which certainly benefit from different preoperative treatments. 

Response: We sincerely appreciate Reviewer #2 for providing this valuable comment. It is true that our study's target population includes elective laparoscopic cholecystectomy patients with various indications, which was intended to enhance the study's applicability. However, we acknowledge the importance of differentiating between the specific pathologies for a more focused analysis.

 To address this concern, we conducted a sensitivity analysis to include only patients with symptomatic gallstones, excluding other pathologies from the analysis. By doing so, we aimed to narrow down the scope and refine our observations for this particular subgroup.

 The message regarding the sensitivity analysis has been incorporated as follows: 

 (From line 18, p 10 in the Material and Methods)

 “Finally, sensitivity analysis was conducted to investigate outcome variability according to alteration of LC indications and the presence of additional IOC.”

 (From line 12, p 15 in the Results)

 “Sensitivity analyses were conducted to explore the impact of more homogeneous data on the main outcome (required treatment); the results are summarized in S2 Table. When only patients with symptomatic gallstones were included (excluding other indications for LC, n=295), and when only elective LC without additional procedures was examined (excluding patients who underwent IOC, n=377), the findings were consistent with the full model. In both sensitivity analyses, the presence of comorbidities was the only significant factor associated with the need for preoperative treatment. The odds of requiring preoperative treatment increased with the number of comorbidities: odds ratios (95% confidence intervals) were 11.0 (2.2, 54.1), 35.2 (6.4, 192.6), and 110.0 (13.5, 896.9) for one, two, and three comorbidities, respectively, in the model with symptomatic gallstones as the sole indication. Similarly, in the model without additional IOC, the odds ratios (95% confidence intervals) were 8.6 (2.3, 32.6), 28.1 (6.8, 116.4), and 124.2 (18.9, 814.6) for one, two, and three comorbidities, respectively. Other factors were not found to be associated with preoperative treatment in either of the sensitivity analytic models.” 

 We added S2 Table to detail the sensitivity analysis result.

 Finally, to express that we included patients with various pathologies to ensure the study's applicability across different clinical scenarios, we have added sentences (From line 20, p 6 in the Material and Methods) “We maintained data heterogeneity to address the diverse indications commonly seen in elective LC, ensuring the relevance and applicability of our study.”

Comment 2 I would also like to understand if these patients underwent outpatient preoperative tests (blood tests, electrocardiogram, chest X-ray, anaesthesiology visit), or all of this is performed the day before or the day of the operation. These exams would certainly help to differentiate the different ASAs and consequently the preoperative preparation.

Response: We sincerely appreciate Reviewer #2 for providing this valuable comment, and we apologize for the missing detail in our initial manuscript. We have taken your suggestion into careful consideration and have now added the necessary message to address this issue.

 In the Material and Methods section, specifically from line 11, page 7, we have included the following detailed description of our Preoperative Anesthesiology Clinic (PAC) organization and the preoperative tests conducted:

"PAC visits were conducted for all patients undergoing elective surgeries at our hospital as outpatient consultations and took place on the same day if surgeons and patients agreed to proceed with planned surgeries. Anesthesiologists, along with nurse anesthetists, conducted all pre-anesthetic evaluations. Additional consultations with subspecialists could be arranged if the anesthesiologist required more specific evaluations. Occasionally, the absence of PAC visits in our hospital could occur owing to human error, either on the part of patients or the administrative staff. To ensure comprehensive assessments, in-patient preoperative anesthesiology visits and checklists were routinely completed on the day of preoperative admission. This allowed for further patient evaluation and the execution of necessary treatments. In cases where patients missed PAC visits, they would still undergo in-patient preoperative anesthesiology visits to prevent cancellations. However, surgical cancellations could still occur if patients were deemed unfit for the planned operations. All surgeries had to be approved by the anesthesiologists before the operation could be performed. Preoperative screening tests were conducted during both the PAC visits and in-patient preoperative anesthesiology visits. Complete blood count and chest radiography were performed for all patients. Patients over 45 years old underwent electrocardiogram testing, and those over 60 years old had blood urea nitrogen, creatinine, and electrolyte levels further assessed. Additional laboratory tests were organized based on specific comorbidities, such as blood sugar testing for diabetes mellitus and liver function tests for suspected liver diseases."

Comment 3 The fact of correcting the potassium seems a confusing data, as it resulted in the statistical analysis as an over-treatment.

Response: We thank Reviewer #2 very much for providing this valuable comment. You are absolutely right that the proper correction of potassium levels can vary among different practitioners. In our analysis, we made efforts to address this issue by having two evaluators, a surgeon, and an anesthesiologist, review the potassium correction for patients, with additional internal medicine consultation in some cases.

 However, we acknowledge that there might still be variability in this aspect, which is an important limitation of our study. In the Discussion section, starting from line 25, page 20, we have now added the following statement to highlight this limitation: "Hypokalemia, which was the most prevalent case of over-treatments in our data, also varied in terms of the preoperative correction threshold [38], even among the anesthesiologists at our hospital."

Comment 4 The sample size is rather small, but it is not clear how many beds the hospitals in which the patients were hospitalized and operated on have.

Response: We extend our gratitude to Reviewer #2 for this valuable comment. In the Material and Methods section, we have mentioned the study setting, which took place in a 700-bed hospital (Line 12, p 5 in the Material and Methods). However, due to the hospital's continuous expansion beyond its official capacity, we faced challenges in providing precise numbers for the total beds in the hospitals where the patients were hospitalized and underwent surgery.

 To address this limitation, we have added the following sentence in the Discussion section (starting from line 8, page 18) to emphasize the presence of extra beds in a 700-bed hospital: "In general, hospital beds cannot be limited according to policies concerning government hospitals." 

Responses to the comments of Reviewer #3

We would like to extend our appreciation to Reviewer #3 for providing us with detailed and invaluable comments. Your constructive comments have been invaluable in enhancing the robustness of our study. Each comment has been thoroughly considered, and we have made the necessary revisions based on your suggestions. Here is a summary of how we have addressed each comment:

Comment 1 This is a retrospective cohort study looking at the value of preoperative admission for laparoscopic cholecystectomy. The authors concluded that 87% of preoperative admissions provided no benefit and were therefore unnecessary. It is not clear what this is based on. Are the authors saying that 87% of patients had no procedures or treatments done during their pre-surgery day in the hospital that needed to be done on that day, or that needed to be done while they were admitted. In other words, only 13% of the patients had procedures done that needed to be done the day before surgery during a pre-surgery admission day. Please state this more clearly.

Response: We sincerely appreciate the valuable comment provided by Reviewer 3, and we apologize for the previous lack of clarity in our message. In response to this comment, we have revised the relevant section by removing the previous sentence "Most (86.9 87.9%) preoperative admissions for laparoscopic cholecystectomy provided no treatment benefit." (From line 23, p 2, in Abstract's conclusion) and replaced it with the following revised statement: " In conclusion, on the basis of our data, a small proportion (12.1%) of patients undergoing elective laparoscopic cholecystectomy may require preoperative admissions to receive the necessary treatment, and most (87.9%) preoperative admissions may not provide treatment benefit. " (From line 24, p 2, in Abstract's conclusion)

Comment 2&3 In the Introduction, the authors are trying to explain what is being reviewed by the 2 raters, but this section is unclear. The category they call ‘documented treatment provided’ sounds like it should be called ‘necessary treatment’, meaning that the procedure was needed, based on the patient’s presenting condition. It should be stated here that each treatment is being rated by the 2 raters, so the unit of analysis is treatments, not patients. However, on P 14 it appears that patients is the unit of analysis, when the authors discuss having 75 patients of whom 38 had potassium correction. Please make it more clear how the outcome variables were calculated. Since some treatments a patient received may have been necessary, while others may have been unnecessary, I’m not sure how patients could be used as the unit of analysis, since this would require that each patient be labeled as either receiving necessary treatment during their pre-operative day in the hospital, or not receiving necessary treatment. What about patients who received both?

 And

 Please re-write this paragraph (it is extremely confusing): “The final categorization of preoperative treatment was summarized into documented treatment provided and required treatment. Required treatment refers to documented treatment provided, excluding overtreatment and including under-treatment. Treatment that could be given as an outpatient visit was required treatment that could be provided as an outpatient. All necessary preoperative cardiovascular monitoring was also considered preoperative treatment (S1 Table).”

Response: We express our sincere gratitude to Reviewer 3 for providing these valuable comments, and we apologize for any confusion caused by our initial message. We appreciate the opportunity to address these comments together, as they both pertain to our main outcome criteria. Our thorough review has resulted in the following responses to each comment:

 1.) We made revisions to the specified section as follows: We removed the confusing sentences shown by strikethrough font [example] (From line 8, p 8, in Material and Methods) and rephrased the passage as follows: 

 (From line 18, p 8, in Material and Methods)

 “Based on the described process, preoperative treatment was classified into three categories: actual treatment rendered, over-treatment, and under-treatment. Subsequently, the category of required treatment, representing the necessary treatment for patients, was determined based on the aforementioned categories. The details of each category are as follows: 

 - Actual treatment rendered referred to the treatments that were provided and documented in the patient's records.

 - Over-treatment denoted treatment that should not have been given.

 - Under-treatment referred to treatment that was necessary but not given. 

 - Required treatment referred to actual treatment rendered, excluding cases of over-treatment and including cases of under-treatment after exclusion.”

 Notably, we have made a change in the wording from "documented treatment provided" to "actual treatment rendered" throughout the entire manuscript, including the Tables.

 2.) Regarding the revised preoperative treatment categories, we made changes to enhance clarity and focus on the main study outcome, which is the necessity of admission for preoperative treatment. The sentence "Treatment that could be given as an outpatient visit was required treatment that could be provided as an outpatient" was removed from the Material and Methods section (From line 16, p 8) to avoid confusion and to streamline the categories.

 Furthermore, we deleted the category "Treatment that could be given as an outpatient visit" from Table 2 since it has less impact on the main study outcome and may not be directly relevant to the necessity of preoperative admission. 3.) To address our main objective of evaluating the necessity of preoperative admission, we conducted the analysis at the individual patient level. This approach allows us to determine which patients require admission and necessary treatment. We apologize for any previous lack of clarity. We have added a statement to clarify that each patient was treated as an individual unit for evaluation:

 (From line 3, p 9, in Material and Methods)

 “In our analysis, each patient was treated as an individual unit for evaluation. Including under-treatment after excluding over-treatment was intended to prevent the exclusion of patients who experienced both under-treatment and over-treatment. Patients who received both actual treatment rendered and over-treatment were classified as not requiring the treatment (no required treatment). Conversely, patients who received actual treatment but had under-treatment, indicating inadequate treatment, were classified as requiring treatment. Only surgery cancellations owing to medical reasons [19] were considered required treatment. The same criteria used to evaluate preoperative treatment were applied to patients with other reasons for cancellation. All necessary preoperative cardiovascular monitoring was also considered preoperative treatment (S1 Table).” 

 These revisions also address the valuable suggestion made by the editor regarding the inclusion of the surgery cancellation parameter in our study.

Comment 4 This is unclear: “The secondary outcomes of the study were postoperative morbidity, mortality within 30 days, and length of hospital stay (LOHS).”

Response: We thank Reviewer 3 for this valuable comment. The message has been revised as follows: " The secondary outcomes of the study were postoperative morbidity and mortality that occurred within 30 days after surgery and the length of hospital stay (LOHS)." (From line 14, p 9, in Material and Methods).

Comment 5 Length of hospital stay is likely to be very highly positively skewed, so it can’t be used as the dependent variable in multiple regression unless it is transformed to make it more normally distributed. The authors should try a log10 transformation, but this may still not correct all the outliers. If there are a few very long admissions, these few patients will have too large an influence on the results. One option might be to recode length of stay into deciles (coded 1-10) to use in linear regression, or consider time to event analysis (where discharge is the event), or consider Poisson regression (which has many complications), or possibly dichotomize this and use logistic regression.

Response: This comment is greatly appreciated, and we would like to express our gratitude to Reviewer 3. We conducted a log10 transformation as suggested, but it was unable to correct all the outliers. After careful consideration, we made the decision to dichotomize the length of hospital stay. This also allowed us to investigate the factors associated with prolonged hospital stays. In response to this comment, we made the following revisions:

 1.) We added a statement regarding the dichotomization of the length of hospital stay outcome and its rationale as follows:

 (From line 19, p 9, in Material and Methods).

 “Considering the expected positive skewness of LOHS and the focus of our study on evaluating the feasibility of daycare surgery, we categorized LOHS into two groups: ≤48 hours and >48 hours. LOHS exceeding 48 hours was considered a prolonged stay, equivalent to a postoperative stay of more than 24 hours [20].” Reference No. 20 has also been added accordingly.

 2.) Message regarding linear regression has been deleted and revised: 

 (From line 9, p 10, in Material and Methods).

 “Multivariable analysis was performed to incorporate multiple potential predictors, with logistic regression used for the preoperative treatment and complication outcomes and linear regression used for LOHS.” has been changed to “Multivariable analysis was performed using logistic regression to examine the associations between multiple potential predictors and preoperative treatment, complication outcomes, and prolonged LOHS (>48 hours).” 

 (From line 16, p 10, in Material and Methods).

 The message “The linear regression models, on the other hand, were evaluated for heteroskedasticity using Breusch–Pagan/Cook–Weisberg test [15].” was deleted.

 3.) We added results in Table 1 “Prolonged hospital stays (> 48 hours): 110 (27.2)” (Table 1 in Results)

 Contents in Table 3 has been revised according to editor and other reviewer’s comment, and regarding dichotomized the length of hospital stay into the outcome “Prolonged hospital stays (> 48 hours)” (Table 3 in Results)

 4.) The sentences and results regarding “Prolonged hospital stays” has been added or revised in the Results’ text: 

 (From line 2, p 12, in Results).

 “A total of 110 patients (27.2%) experienced a prolonged LOHS (>48 hours).” has been added

 (From line 31, p 14, in Results).

 The message regarding previous “Length of hospital stay” has been changed to “prolonged LOHS (>48 hours)” as followed. 

 “The median hospital stay was 3 (interquartile range: 2–4) days (Table 3). Regarding Table 3, factors that predicted an increase in LOHS prolonged LOHS (>48 hours) included the presence of additional procedures IOC, surgical indications other than symptomatic gallstones, patients requiring preoperative treatment, and the occurrence of complications.” 

 (From line 35, p 14, in Results).

 As female variable was no longer significant related reduce hospital stays to and linear regression related sentences were no longer applicable. The related messages have been deleted: 

 “The only factor related to a shorter LOHS was female sex. We analyzed the LOHS regression results using a robust linear regression model because we found evidence of heteroskedasticity (p-value < 0.001) from Breusch-Pagan/Cook-Weisberg testing for traditional linear regression. A robust linear regression model is less sensitive to heteroskedasticity and outliers in the data, and thus provides a more reliable estimate of the relationship between the predictor and response variables [17].”

 (From line 6, p 15, in Results).

 Message regarding goodness of fit was revised by adding the result from dichotomized hospital stay outcome: 

 “A goodness of fit test was conducted for two logistic regression models: the required treatment model and the medical complication model. The required treatment model had a p-value of 0.810 0.609, and the medical complication model had a p-value of 0.181 0.333, and the prolonged hospital stays model had a p-value of 0.150. These results suggest that both all models were a good fit for the data, as the numbers of occurrences were not significantly different from those predicted by the models [13] [21].”

Comment 6 Given that providers are accustomed to having a day of pre-surgery admission, it is likely that outcomes will not be as good as the authors expect if there is a sudden switch and they eliminate this pre-surgery admission. Hospital processes will need to be established based on same-day surgery that do not currently exist.

Response: We greatly appreciate the valuable comment provided by Reviewer 3. We understand that changing routine protocols can be challenging, especially when introducing a non-existent practice. Moreover, same-day surgery for laparoscopic cholecystectomy is not yet established in our country. However, the findings of our study have highlighted the inefficiency of the conventional practice and prompted us to revise our surgery protocol. One significant change we implemented was the replacement of preoperative admission with a preoperative anesthetic clinic (PAC) for patient evaluation. In response to your and the editor's suggestion, we have included a statement addressing this change:

(From line 8, p 20, in Discussion)

“The study findings have had a significant impact on the implementation of a daycare surgery protocol for LC at the study hospital. One effective approach involved carefully selecting patients to eliminate the need for preoperative admission and instead implementing an efficient PAC. As a result, over the past year, the hospital has successfully performed approximately 100 daycare LC procedures. These achievements have been recognized at the national level; the hospital received the Best Practice Award from the Thailand One day surgery and Minimally Invasive Surgery national forum in 2022.”

---

## [Decision Letter · Decision Letter 1]

17 Aug 2023

PONE-D-23-13647R1Preoperative admission is non-essential in most patients receiving elective laparoscopic cholecystectomy: a cohort studyPLOS ONE

Dear Dr. Tunruttanakul,

Thank you for submitting your manuscript to PLOS ONE. After careful consideration, we feel that it has merit but does not fully meet PLOS ONE’s publication criteria as it currently stands. Therefore, we invite you to submit a revised version of the manuscript that addresses the points raised during the review process.

I thank the authors for their effort in making the suggested changes to their study.

The result is good, but in agreement with the reviewer 3 I think some further minor adjustments are needed (Beginning with point 2 and 3 proposed by reviewer 3).

Since the aim of the paper is to change, in a precise territorial area, the current attitude regarding the treatment of gallbladder lithiasis and to increase the use/optimization of day surgery, it is imperative that the definitions and indications for treatments are clear and precise, as well as the criteria to perform preoperative investigations must be well defined.

Complications according to which classification were classified (Clavie-Dindo? Other?).

As the reviewer 3 also suggested, it is necessary to better clarify (probably a translation problem ) the concept of "required treatment" and "actual treatment rendered."

Regarding reference item 31 (in the discussion) is misplaced , in fact “The chapter focuses on perioperative assessment and anesthetic considerations of the trauma patient specially focused on difficult airway rescue and management by the various intubation techniques, precautions during extubation, and postoperative care “

Bibliographic entry 32 is also very irrelevant, in fact it refers to 58 patients undergoing general surgery out of 499 examined who did not undergo pre-operative testing. Nor is the procedure performed specified.

I also request an extensive linguistic revision of the text, which is difficult to understand in some passages.

I ask that in the introduction the first sentence (defining gallbladder lithiasis) and in lines 3-4 the definition of cholecystectomy (which should be known to all surgeons) be removed.

In the materials and methods page 5 lines 24-27 the choice of cut off for age is not clearly defined.

I read reference 14 in lancet but did not find the grouping you propose. Is yours an adaptation from the original?

Page 7 lines 1-5 it is necessary to define the indication to IOC.

Page 7 line 25 : define "complete blood count"

Page 7 line 26: Why EKG only to those over 45? According to what guidelines was this choice made?

P8 lines 1-2. According to which guidelines is it correct to submit a patient who has not had liver function tests to cholecystectomy? Some tests may raise suspicion of choledochal lithiasis or the presence of other pathologies that may not be immediately understood postoperatively.

Page 8 lines 19-26 and page 9 lines 1-5 the ideas expressed need to be rewritten more clearly.

What does the one-month follow-up program include: an outpatient visit? The execution of blood tests? An ultrasound examination? Anything else?

In the results section, page 11 line 30. The 798 cancellations refer to what type of procedures? If they are not cholecystectomies why provide this figure?

What is meant by "cardiovascular instability monitoring"?

We look forward to receiving your revised manuscript.

Kind regards,

Fabrizio D'Acapito, Ph.D,M.D.

Academic Editor

PLOS ONE

Journal Requirements:

Additional Editor Comments (if provided):

I thank the authors for their effort in making the suggested changes to their study.

The result is good, but in agreement with the reviewer 3 I think some further minor adjustments are needed (Beginning with point 2 and 3 proposed by reviewer 3).

Since the aim of the paper is to change, in a precise territorial area, the current attitude regarding the treatment of gallbladder lithiasis and to increase the use/optimization of day surgery, it is imperative that the definitions and indications for treatments are clear and precise, as well as the criteria to perform preoperative investigations must be well defined.

Complications according to which classification were classified (Clavie-Dindo? Other?).

As the reviewer 3 also suggested, it is necessary to better clarify (probably a translation problem ) the concept of "required treatment" and "actual treatment rendered."

Regarding reference item 31 (in the discussion) is misplaced , in fact “The chapter focuses on perioperative assessment and anesthetic considerations of the trauma patient specially focused on difficult airway rescue and management by the various intubation techniques, precautions during extubation, and postoperative care “

Bibliographic entry 32 is also very irrelevant, in fact it refers to 58 patients undergoing general surgery out of 499 examined who did not undergo pre-operative testing. Nor is the procedure performed specified.

I also request an extensive linguistic revision of the text, which is difficult to understand in some passages.

I ask that in the introduction the first sentence (defining gallbladder lithiasis) and in lines 3-4 the definition of cholecystectomy (which should be known to all surgeons) be removed.

In the materials and methods page 5 lines 24-27 the choice of cut off for age is not clearly defined.

I read reference 14 in lancet but did not find the grouping you propose. Is yours an adaptation from the original?

Page 7 lines 1-5 it is necessary to define the indication to IOC.

Page 7 line 25 : define "complete blood count"

Page 7 line 26: Why EKG only to those over 45? According to what guidelines was this choice made?

P8 lines 1-2. According to which guidelines is it correct to submit a patient who has not had liver function tests to cholecystectomy? Some tests may raise suspicion of choledochal lithiasis or the presence of other pathologies that may not be immediately understood postoperatively.

Page 8 lines 19-26 and page 9 lines 1-5 the ideas expressed need to be rewritten more clearly.

What does the one-month follow-up program include: an outpatient visit? The execution of blood tests? An ultrasound examination? Anything else?

In the results section, page 11 line 30. The 798 cancellations refer to what type of procedures? If they are not cholecystectomies why provide this figure?

What is meant by "cardiovascular instability monitoring"?

Reviewers' comments:

Reviewer's Responses to Questions

**Comments to the Author**

1. If the authors have adequately addressed your comments raised in a previous round of review and you feel that this manuscript is now acceptable for publication, you may indicate that here to bypass the “Comments to the Author” section, enter your conflict of interest statement in the “Confidential to Editor” section, and submit your "Accept" recommendation.

Reviewer #1: All comments have been addressed

Reviewer #3: (No Response)

2. Is the manuscript technically sound, and do the data support the conclusions?

Reviewer #1: Yes

Reviewer #3: Partly

3. Has the statistical analysis been performed appropriately and rigorously? 

Reviewer #1: Yes

Reviewer #3: Yes

4. Have the authors made all data underlying the findings in their manuscript fully available?

Reviewer #1: Yes

Reviewer #3: Yes

5. Is the manuscript presented in an intelligible fashion and written in standard English?

Reviewer #1: Yes

Reviewer #3: No

6. Review Comments to the Author

Reviewer #1: (No Response)

Reviewer #3: 1. One additional Limitation that should be mentioned in the Discussion is that the people who rated treatments as necessary or unnecessary were aware of the study hypotheses, so they may have been biased in their ratings.

2. This may be more an issue with the translation than the underlying concepts being used, but I still have a problem understanding the grouping and the coding of the primary outcome variable. This sentence does not make sense to me: “ none of the patients with ASA status I received both actual treatment rendered and the required treatment.“ I don’t understand how a patient could have not received the actual treatment rendered: if it was actual and rendered, then it must have been received. What am I missing? The entire definition of the outcome variable is similar: the words do not make sense.

3. This paragraph is confusing: “A goodness of fit test was conducted for two logistic regression models: the required treatment model and the medical complication model. The required treatment model had a p-value of 0.810 0.609, and the medical complication model had a p-value of 0.181 0.333, and the prolonged hospital stays model had a p-value of 0.150. These results suggest that both all models were a good fit for the data, as the numbers of occurrences were not significantly different from those predicted by the models [13] [21]” Akaike’s Information Criterion (AIC) is often used to evaluate goodness of fit to the data, but it is generally compared between nested models without using a p value. It is not clear what kind of ‘goodness of fit’ test was used, why it produces several p values per model, and what a non-significant p value for this test means. Perhaps they are referring to model calibration: the agreement between predicted and observed incidence of the outcome across a range of predicted probabilities, often grouped into deciles. This is sometimes tested using the Hosmer-Lemeshow test, where a non-significant finding would mean good calibration. If that is true, please state this.

7. PLOS authors have the option to publish the peer review history of their article (what does this mean?). If published, this will include your full peer review and any attached files.

Reviewer #1: No

Reviewer #3: No

---

## [Author Response · Author response to Decision Letter 1]

22 Sep 2023

Responses to the Editor Comments

We thank again to to Dr. Fabrizio D'Acapito for providing us with detailed and valuable comments. Your valuable input has been instrumental in improving our research. We have taken each comment into careful consideration and made the necessary revisions accordingly. Here is a summary of how we addressed each comment: (the specified page and line are according to the “Revised manuscript with Track Changes” file) 

Comment 1. I thank the authors for their effort in making the suggested changes to their study.

The result is good, but in agreement with the reviewer 3 I think some further minor adjustments are needed (Beginning with point 2 and 3 proposed by reviewer 3).

Since the aim of the paper is to change, in a precise territorial area, the current attitude regarding the treatment of gallbladder lithiasis and to increase the use/optimization of day surgery, it is imperative that the definitions and indications for treatments are clear and precise, as well as the criteria to perform preoperative investigations must be well defined.

Response: We appreciate the editor's feedback. In this updated version, we have attempted to incorporate your and reviewer 3's suggestions. We hope that these revisions adequately address the comment.

Comment 2. As the reviewer 3 also suggested, it is necessary to better clarify (probably a translation problem) the concept of "required treatment" and "actual treatment rendered."

Response: We appreciate the editor's comment. Upon careful review, we fully acknowledge and agree with your and reviewer 3's concerns. Consequently, we sought guidance from our language editor, resulting in the following changes: we replaced "required treatment" with "treatment necessary," and "actual treatment rendered" with "treatment received." These modifications have been consistently applied throughout the text, and we have included additional clarifying messages as follow:

 The definitions of the terms have been updated from page 8, line 5 to page 9, line 6, as: “The primary objective of this study was to evaluate the treatment and monitoring received by patients during their preoperative admission, excluding care provided on the day of surgery. Two participating specialists, a surgeon and an anesthesiologist, separately reviewed the patient data and later held an in-person meeting to review their findings. In cases of inconclusive results, a subspecialist in internal medicine was consulted. The criteria for necessary preoperative treatment are outlined in S1 Table. Based on the described process, preoperative treatment was classified into three categories: “treatment received”, over-treatment, and under-treatment. Subsequently, the category of “treatment necessary”, representing necessary treatment for patients, was determined based on the aforementioned categories. The details of each category are as follows: 

 - “Treatment received” referred to treatments that were provided and documented in the patient's records.

 - Over-treatment denoted treatment that should not have been given.

 - Under-treatment referred to treatment that was necessary but not given. 

 - “Treatment necessary” referred to “treatment received”, excluding cases of over-treatment and including cases of under-treatment after exclusion. 

In our analysis, each patient was treated as an individual unit for evaluation. Including under-treatment after excluding over-treatment was intended to prevent the exclusion of patients who experienced both under-treatment and over-treatment. "Treatment necessary" comprised all patients who received the preoperative treatment (i.e., the "treatment received" group), as well as those who required treatment but did not receive it or received insufficient treatment (i.e., the under-treatment group). However, "treatment necessary" excluded patients in the over-treatment group.”

Comment 3. Regarding reference item 31 (in the discussion) is misplaced , in fact “The chapter focuses on perioperative assessment and anesthetic considerations of the trauma patient specially focused on difficult airway rescue and management by the various intubation techniques, precautions during extubation, and postoperative care “

Bibliographic entry 32 is also very irrelevant, in fact it refers to 58 patients undergoing general surgery out of 499 examined who did not undergo pre-operative testing. Nor is the procedure performed specified.

Response: We sincerely appreciate the editor's insightful comments and apologize for any oversight in reference selection. We have thoroughly reevaluated the references and incorporated more pertinent ones. Specifically, we have updated the passage to (page 19, line 16 to 20) "Recent guidelines from the United Kingdom (National Institute for Health and Care Excellence) have shifted away from recommending routine preoperative tests [32]. Moreover, these tests (if done routinely) have the potential to cause harm and incur unnecessary costs without significantly altering preoperative management [33]. "

The replaced references are as follows:

32. National Guideline Centre (UK). Preoperative Tests (Update): Routine Preoperative Tests for Elective Surgery. London: National Institute for Health and Care Excellence (NICE); 2016 Apr. (NICE Guideline, No. 45.) 1, Guideline summary.

33. Martin SK, Cifu AS. Routine Preoperative Laboratory Tests for Elective Surgery. JAMA. 2017;318(6):567-8.

Group of relevant comments 

Complications according to which classification were classified (Clavie-Dindo? Other?).

I also request an extensive linguistic revision of the text, which is difficult to understand in some passages.

I ask that in the introduction the first sentence (defining gallbladder lithiasis) and in lines 3-4 the definition of cholecystectomy (which should be known to all surgeons) be removed.

In the materials and methods page 5 lines 24-27 the choice of cut off for age is not clearly defined.

I read reference 14 in lancet but did not find the grouping you propose. Is yours an adaptation from the original?

Page 7 lines 1-5 it is necessary to define the indication to IOC.

Page 7 line 25 : define "complete blood count"

Page 7 line 26: Why EKG only to those over 45? According to what guidelines was this choice made?

P8 lines 1-2. According to which guidelines is it correct to submit a patient who has not had liver function tests to cholecystectomy? Some tests may raise suspicion of choledochal lithiasis or the presence of other pathologies that may not be immediately understood postoperatively.

Page 8 lines 19-26 and page 9 lines 1-5 the ideas expressed need to be rewritten more clearly.

What does the one-month follow-up program include: an outpatient visit? The execution of blood tests? An ultrasound examination? Anything else?

In the results section, page 11 line 30. The 798 cancellations refer to what type of procedures? If they are not cholecystectomies why provide this figure?

What is meant by "cardiovascular instability monitoring"?

Response: We sincerely appreciate the editor's meticulous review and valuable comments. Your thoughtful evaluation significantly improved our manuscript. As these comments revolve around enhancing clarity of expression, we will collectively address them for better coherence.

 1. I ask that in the introduction the first sentence (defining gallbladder lithiasis) and in lines 3-4 the definition of cholecystectomy (which should be known to all surgeons) be removed.

 Response: We have eliminated the initial paragraph in the introduction and inserted the sentence “Laparoscopic cholecystectomy (LC), the surgical removal of the gallbladder, is a frequently performed procedure [1] used to treat various complications related to gallstones [2].” at the beginning of the second paragraph, which is now the first paragraph in the revised version (page 4, line 9 to 10).

 2. In the materials and methods page 5 lines 24-27 the choice of cut off for age is not clearly defined.

 Response: We apologize for the previous confusion regarding the cut-off. We have revised the sentence as follows (page 5, line 25 to page 6, line 1): “The patient's age was recorded in years and grouped into two categories: below 65 years old and 65 years old or older (the cutoff in the review by Orimo et al. [11]).” Additionally, we have updated the reference to: “Orimo H, Ito H, Suzuki T, Araki A, Hosoi T, Sawabe M. Reviewing the definition of 'elderly'. Geriatrics & Gerontology International. 2006;6(3):149-58.”

 3. I read reference 14 in lancet but did not find the grouping you propose. Is yours an adaptation from the original?

 Response: We sincerely apologize for not thoroughly checking this criterion. We highly appreciate the editor's diligence in pointing out this oversight. The cutoffs have been revised to <27.5, 27.5 to <32.5, and ≥32.5 kg/m2. The terms "class I" and "class II" obesity were removed, given the lack of consensus on cutoffs in Asian countries, including the nomenclature for these criteria. These changes have been implemented throughout the entire manuscript. Furthermore, we have recalculated and revised the results in the text and tables. Some sentences were removed accordingly. Below is a detailed list of the changes made:

 1). Page 6, line 1 to 3 in Material and Methods: “BMI was recorded in kg/m2 and divided into three categories based on the Asian obesity cutoff value for public health action: <27.5 kg/m2, 27.5 to <32.5, and ≥32.5 kg/m2 [12].”

 2). Page 12, line 6 to 9 in Results: “Most patients were women (77.0%); 25.7% (104 patients) of the total patients were obese with BMI 27.5 to <32.5 kg/m2, and 9.4% (38 patients) were BMI ≥32.5.”

 3). Page 14, line 18 to 19 in Results: Patients' age, “BMI greater than 32.5 kg/m2, presence of IOC, surgical indications, and ASA status were not associated with preoperative treatments.”

 4). Page 15, line 6 to 9 in Results. The sentence “With our sample size, no variable was found to significantly predict medical complications, although class II obesity trended toward requiring both preoperative treatment and increased medical complications (odds ratio and p-value of 2.9 and 0.222, and 9.8 and 0.090, respectively).” has been revised to “With our sample size, no variable was found to significantly predict medical complications.”

 5). In the Discussion, two messages regarding incorrect classifications and their associated results have been removed as follows:

 - Page 18, line 22 to 25: “Class II obesity showed a trend toward higher treatment requirements and medical complications, with an odds ratio of 2.9 (p-value 0.222) and 9.8 (p-value 0.090), respectively. The limited number of class II obese patients (n=10) may have contributed to the non-significance of the results.” has been removed.

 - Page 21, line 8 to 10: “It is possible that the incidence of OSA was low, given the small proportion of obese patients (9.4%, or 38 patients with class I and II obesity) who also have a high likelihood of OSA [39].” has been removed.

 This is because body mass index ≥32.5 kg/m2 no longer trends towards both treatment requirements and the occurrence of medical complications. Our incidence of obesity, considering the revised cut-off, is also substantial, up to 35.1% (body mass index ≥27.5 kg/m2)." 

 6) In Tables 1, 2, 3, and S1: All criteria have been revised accordingly, necessitating a comprehensive re-calculation and adjustment of all results. Consequently, numerical results must be revised throughout the manuscript, including the abstract. Please look for red highlights for additions and use strikethrough font for deletions."

 4. Page 7 lines 1-5 it is necessary to define the indication to IOC.

 Response: We appreciate the editor's feedback. We have provided additional details regarding the indications for intraoperative cholangiography (IOC). The updated statement is as follows (page 6, line 21 to page 7, line 2): “Regarding the presence of IOC parameters, in our department, selective IOC was indicated for patients with symptomatic gallstones who had a relatively low but non-negligible risk of choledocholithiasis. This included patients with a clinical history of jaundice, pancreatitis, abnormal liver function test results, bile duct dilatation detected in imaging studies, or the patients at intermediate risk for choledocholithiasis according to guidelines [14]. In cases with a greater suspicion of choledocholithiasis (high-risk for choledocholithiasis, which includes clinical cholangitis or choledocholithiasis detected in imaging studies), alternative modalities such as endoscopic retrograde cholangiography were used [14].” The reference has been changed from 'ASGE guidelines' to 'ESGE guidelines' as it is more coherent with our protocol.

 5. Page 7 line 25: define "complete blood count"

 Response: We appreciate the editor's comment. The phrasing has been updated to (page 7, line 21 to 23): “Chest radiography and a complete blood count (blood test that evaluates the total numbers and characteristics of cell components in a patient's blood, including red and white blood cells and platelets) were performed for all patients.”

 6. Page 7 line 26: Why EKG only to those over 45? According to what guidelines was this choice made?

 Response: We appreciate the editor's feedback and apologize for any confusion. The criteria details have been updated to specify (page 7, line 25 to page 8, line 1): “Our hospital's preoperative screening test protocol with respect to patients’ age (i.e., blood tests, chest radiography, and electrocardiogram testing) has been adopted from the practice advisory for pre-anesthesia evaluation by the Royal College of Anesthesiologists of Thailand [17].”

 The reference 'No 17' pertaining to the guideline has been supplemented with the relevant information: “17. The Royal College of Anesthesiologists of Thailand. Practice advisory for preanesthesia evaluation. 2019. Available from: https://www.rcat.org/_files/ugd/82246c_6386a015c1574075a50dff87cfc2b060.pdf”

 7. P8 lines 1-2. According to which guidelines is it correct to submit a patient who has not had liver function tests to cholecystectomy? Some tests may raise suspicion of choledochal lithiasis or the presence of other pathologies that may not be immediately understood postoperatively.

 Response: We thank the editor for this insightful comment, and we deeply regret the oversight. Your feedback is highly valuable. We have revised the message as follows (page 8, line 1 to 4): “Additional laboratory tests were organized based on specific comorbidities, such as blood sugar testing for diabetes mellitus. Liver function tests were conducted for all patients undergoing LC.”

 8. Page 8 lines 19-26 and page 9 lines 1-5 the ideas expressed need to be rewritten more clearly.

 What does the one-month follow-up program include: an outpatient visit? The execution of blood tests? An ultrasound examination? Anything else?

 Complications according to which classification were classified (Clavie-Dindo? Other?).

 Response: We express our sincere gratitude to the editor for providing these valuable comments. Since they fall within the same context, we will address them collectively. 

 The classification of complications was updated according to the Clavien-Dindo criteria, as per your suggestion. We have included this information in the Material and Methods section, along with the relevant reference (page 9, line 21 to 26): “Complications were classified according to Clavien–Dindo classification. In brief, Grade I indicates any deviation from the normal postoperative course not requiring treatment; Grade II involves the need for pharmacological treatment; Grade III requires surgical, endoscopic, or radiological intervention; Grade IV denotes life-threatening complications, and Grade V signifies patient death [20].” Additionally, we have revised the discussion of complications in the Results section (page 14, line 20 to page 15, line 6): “Complications occurred in 17 (4.2%) patients, with no mortality; 58.8% of complications were surgery-related (10 patients, 2.8%), including two cases of major morbidity (grade III, bile duct injury requiring surgical reconstruction). Medical complications were experienced by nine patients (2.2%), with two severe cases (grade IV, 0.5%) including one cerebrovascular infarction and one readmission for lobar pneumonia with respiratory failure. Five patients had both grade II surgical and medical complications. Three patients (0.7%) experienced retained common bile duct stones.” 

 We have included additional details about the follow-up program in the Material and Methods section as follows (page 10, line 10 to 13): “Our surgical department routinely conducts a 1-month follow-up. Assessment of clinical symptoms, physical examinations, and reviews of histologic reports are performed at outpatient visits during follow-up for patients undergoing LC. Additionally, liver function tests are used to monitor patients undergoing LC and IOC.”

 The unclear messages have been re-edited for language as per your suggestion. The modifications now span from page 9, line 20 to page 10, line 16. 

 9. In the results section, page 11 line 30. The 798 cancellations refer to what type of procedures? If they are not cholecystectomies why provide this figure?

 Response: We appreciate the editor's feedback and apologize for including unrelated details. Following your suggestion, the unrelated message has been removed, as shown (page 12, line 12 to 14): “Surgery cancellations related to the study protocol were observed in two cases (0.5%).”

 10. What is meant by "cardiovascular instability monitoring"?

 Response: We thank the editor again for this comment. The detail regarding "cardiovascular instability monitoring" has been added in the Material and Method (page 9, line 9 to19) as: “All necessary preoperative cardiovascular instability monitoring was also considered preoperative treatment (S1 Table). Monitoring preoperative cardiovascular instability is a standard protocol at our hospital for high-risk non-cardiovascular surgical patients. On the preoperative day, anesthesiologists require an extended monitoring period, which includes non-invasive blood pressure and vital sign measurement, continuous electrocardiographic monitoring, and peripheral pulse oximetry measurements [19]. Additionally, in-patient preoperative anesthesiology visits, checklists, and a review of PAC assessments are conducted. Additional monitoring during the operative and postoperative periods is also implemented but beyond the scope of this study. This process is designed to enhance the safety of surgery for patients deemed at high risk for perioperative complications (S1 Table).” The relating reference has been added (19. Aseni P, Orsenigo S, Storti E, Pulici M, Arlati S. Current concepts of perioperative monitoring in high-risk surgical patients: a review. Patient Saf Surg. 2019;13:32.)

Editing by Authors in Addition to Editor or Reviewer Comments

In the abstract (page 2, line 20 to 21), the sentence “The cohort had 4.2% morbidity (2.2% severe medical complications), with no mortality.” has been revised to “The cohort had 4.2% morbidity (2.2% medical complications), with no mortality.” by removing 'severe' from medical complications, as the revised version is more accurate regarding the study results."

Responses to the comments of Reviewer #3

We express our gratitude to Reviewer #3 for their detailed and invaluable feedback. Your constructive comments have significantly contributed to enhancing the rigor of our study. We carefully considered each comment and made the necessary revisions accordingly. Here is a summary of the actions taken to address your valuable input: (the specified page and line are according to the “Revised manuscript with Track Changes” file) 

1. One additional Limitation that should be mentioned in the Discussion is that the people who rated treatments as necessary or unnecessary were aware of the study hypotheses, so they may have been biased in their ratings.

Response: We appreciate Reviewer #3 for highlighting this concern, and we completely agree. In response to your suggestion on page 20, line 26 to page 21, line 2, we have included a limitation as follows: “In connection with the second limitation, two participating specialists who assessed the primary outcome were aware of the study's hypotheses. Consequently, there is a possibility of bias in their assessments of whether preoperative treatment was necessary.”

2. This may be more an issue with the translation than the underlying concepts being used, but I still have a problem understanding the grouping and the coding of the primary outcome variable. This sentence does not make sense to me: “none of the patients with ASA status I received both actual treatment rendered and the required treatment. “I don’t understand how a patient could have not received the actual treatment rendered: if it was actual and rendered, then it must have been received. What am I missing? The entire definition of the outcome variable is similar: the words do not make sense.

Response: We appreciate Reviewer #3 for pointing out this concern and apologize for any remaining confusion. We have revised the sentence (page 13, line 22 to 24) as follows: “Notably, in our dataset, patients with ASA status I did not receive preoperative treatment ('treatment received') or need necessary treatment (“treatment necessary”) on the preoperative admission date.”

 To address the confusion surrounding the concepts of “actual treatment rendered” and “required treatment,” we consulted with our language editor and made the following changes: we replaced “required treatment” with “treatment necessary,” and “actual treatment rendered” with “treatment received.” These modifications have been applied consistently throughout the text. Additionally, to provide further clarity, we have included explanatory statements as (page 8, line 5 to page 9, line 6): 

 “The primary objective of this study was to evaluate the treatment and monitoring received by patients during their preoperative admission, excluding care provided on the day of surgery. Two participating specialists, a surgeon and an anesthesiologist, separately reviewed the patient data and later held an in-person meeting to review their findings. In cases of inconclusive results, a subspecialist in internal medicine was consulted. The criteria for necessary preoperative treatment are outlined in S1 Table. Based on the described process, preoperative treatment was classified into three categories: “treatment received”, over-treatment, and under-treatment. Subsequently, the category of “treatment necessary”, representing necessary treatment for patients, was determined based on the aforementioned categories. The details of each category are as follows: 

 - “Treatment received” referred to treatments that were provided and documented in the patient's records.

 - Over-treatment denoted treatment that should not have been given.

 - Under-treatment referred to treatment that was necessary but not given. 

 - “Treatment necessary” referred to “treatment received”, excluding cases of over-treatment and including cases of under-treatment after exclusion. 

In our analysis, each patient was treated as an individual unit for evaluation. Including under-treatment after excluding over-treatment was intended to prevent the exclusion of patients who experienced both under-treatment and over-treatment. "Treatment necessary" comprised all patients who received the preoperative treatment (i.e., the "treatment received" group), as well as those who required treatment but did not receive it or received insufficient treatment (i.e., the under-treatment group). However, "treatment necessary" excluded patients in the over-treatment group.”

3. This paragraph is confusing: “A goodness of fit test was conducted for two logistic regression models: the required treatment model and the medical complication model. The required treatment model had a p-value of 0.810 0.609, and the medical complication model had a p-value of 0.181 0.333, and the prolonged hospital stays model had a p-value of 0.150. These results suggest that both all models were a good fit for the data, as the numbers of occurrences were not significantly different from those predicted by the models [13] [21]” Akaike’s Information Criterion (AIC) is often used to evaluate goodness of fit to the data, but it is generally compared between nested models without using a p value. It is not clear what kind of ‘goodness of fit’ test was used, why it produces several p values per model, and what a non-significant p value for this test means. Perhaps they are referring to model calibration: the agreement between predicted and observed incidence of the outcome across a range of predicted probabilities, often grouped into deciles. This is sometimes tested using the Hosmer-Lemeshow test, where a non-significant finding would mean good calibration. If that is true, please state this.

Response: We greatly appreciate Reviewer #3 for this insightful comment. Your suggestions are consistently clear and accurate. In the previous version of the manuscript, we calibrated our models using the Pearson chi-square test. However, as you correctly pointed out, the Hosmer-Lemeshow test would be more appropriate. In response to this comment, we have made the following revisions to our manuscript:

 1. In the Material and Methods section (page 11, line 3 to 5), the text has been revised to: “The goodness of fit of the logistic regression models was tested using the Hosmer–Lemeshow test. This assessment involves grouping the observations into deciles (groups of 10) based on their predicted probabilities [23].”

 2. In the Results section (page 15, line 14 to 22), we have recalculated the results and revised the text to: “A goodness of fit test (model calibration) was conducted for logistic regression models, assessing the agreement between predicted values from the model and observed values. A non-significant result was interpreted as indicative of good calibration [23]. Regarding models in Table 3, the treatment necessary model had a p-value of 0.221, the medical complication model had a p-value of 0.138, and the prolonged LOHS model had a p-value of 0.126. These results suggest that all models had a proper ability to describe the response variable.”

---

## [Decision Letter · Decision Letter 2]

13 Oct 2023

Preoperative admission is non-essential in most patients receiving elective laparoscopic cholecystectomy: a cohort study

PONE-D-23-13647R2

Dear Dr. Tunruttanakul,

We’re pleased to inform you that your manuscript has been judged scientifically suitable for publication and will be formally accepted for publication once it meets all outstanding technical requirements.

Kind regards,

Fabrizio D'Acapito, Ph.D,M.D.

Academic Editor

PLOS ONE

Additional Editor Comments (optional):

I really appreciate the effort made by the authors to follow the advice given by the reviewers to improve the paper.

I strongly renew the point that the prominence of the submitted paper is determined by the geographical area of origin and the role this paper may have in that country/geographical area.

I point out that data already in the table are repeated in the results.

I believe that the paper can be accepted subject to the implementation of the corrections below, and I request a new language check.

Pag 50 erase “the surgical removal of the gallbladder “

Pag 51 line 26 change to “The patient's age was recorded in years and grouped into two categories: age≤ 65 years old or age> 65 years old”

Pag 51 line 26 change to “ …according to the cut off…”

Pag 52 line 10 change to “ …symptomatic gallstones, prior acute cholecystitis, prior gallstone pancreatitis, and prior choledocholithiasis”

Pag 55 line 22-26 : erase “In brief, Grade I indicates any deviation from the normal postoperative course not requiring treatment; Grade II involves the need for pharmacological treatment; Grade III requires surgical, endoscopic, or radiological intervention; Grade IV denotes life-threatening complications, and Grade V signifies patient death”

P56 line 13 change to “… LC plus IOC…”

P57 results : data in the table should be given in the text only if they are essential or so relevant that they need to be emphasized

P61 line 1 change to “ grade IIIb…”

P 61 lines 4-5 change to “Five patients had grade II complications”

Reviewers' comments:

Reviewer's Responses to Questions

**Comments to the Author**

1. If the authors have adequately addressed your comments raised in a previous round of review and you feel that this manuscript is now acceptable for publication, you may indicate that here to bypass the “Comments to the Author” section, enter your conflict of interest statement in the “Confidential to Editor” section, and submit your "Accept" recommendation.

Reviewer #2: (No Response)

2. Is the manuscript technically sound, and do the data support the conclusions?

Reviewer #2: Yes

3. Has the statistical analysis been performed appropriately and rigorously? 

Reviewer #2: Yes

4. Have the authors made all data underlying the findings in their manuscript fully available?

Reviewer #2: Yes

5. Is the manuscript presented in an intelligible fashion and written in standard English?

Reviewer #2: Yes

6. Review Comments to the Author

Reviewer #2: After the statistical review done, I agree with everything the Authors present, in particular on data collection, given the small number due to the geographical area in wich the work was carried out.

7. PLOS authors have the option to publish the peer review history of their article (what does this mean?). If published, this will include your full peer review and any attached files.

Reviewer #2: No

---

## [Editor Report · Acceptance letter]

17 Oct 2023

PONE-D-23-13647R2 

Preoperative admission is non-essential in most patients receiving elective laparoscopic cholecystectomy: a cohort study 

Dear Dr. Tunruttanakul:

I'm pleased to inform you that your manuscript has been deemed suitable for publication in PLOS ONE. Congratulations! Your manuscript is now with our production department. 

Kind regards, 

on behalf of

Dr. Fabrizio D'Acapito 

Academic Editor

PLOS ONE